# Single-molecule imaging with cell-derived nanovesicles reveals early binding dynamics at a cyclic nucleotide-gated ion channel

Vishal R. Patel [1,2], Arturo M. Salinas[3], Darong Qi[1], Shipra Gupta[1], David J. Sidote[1] & Marcel P. Goldschen-Ohm [1✉]

Ligand binding to membrane proteins is critical for many biological signaling processes. However, individual binding events are rarely directly observed, and their asynchronous dynamics are occluded in ensemble-averaged measures. For membrane proteins, single-molecule approaches that resolve these dynamics are challenged by dysfunction in non-native lipid environments, lack of access to intracellular sites, and costly sample preparation. Here, we introduce an approach combining cell-derived nanovesicles, microfluidics, and single-molecule fluorescence colocalization microscopy to track individual binding events at a cyclic nucleotide-gated TAX-4 ion channel critical for sensory transduction. Our observations reveal dynamics of both nucleotide binding and a subsequent conformational change likely preceding pore opening. Kinetic modeling suggests that binding of the second ligand is either independent of the first ligand or exhibits up to ~10-fold positive binding cooperativity. This approach is broadly applicable to studies of binding dynamics for proteins with extracellular or intracellular domains in native cell membrane.

[1] Department of Neuroscience, The University of Texas at Austin, Austin, TX, USA. [2] Dell Medical School, The University of Texas at Austin, Austin, TX, USA. [3] Department of Physics, The University of Texas at Austin, Austin, TX, USA. ✉email: marcel.goldschen-ohm@austin.utexas.edu

Ligand binding to specific recognition sites in membrane receptors is crucial for cellular signaling and pharmacological treatment of its dysfunction. Intrinsic membrane proteins make up ~30% of the protein-encoding genome and are therapeutic targets for ~70% of available drugs[1–3]. Technological advances have led to an increase in near-atomic resolution structures of membrane receptors complexed with ligands/drugs. These structures provide static snapshots primarily of endpoints for ligand-activation mechanisms. However, the transient intermediate events connecting these snapshots often remain unclear. For oligomeric proteins that bind ligand/drug at multiple active sites, these intermediates define the sequence of binding events and the nature of cooperative interactions amongst sites (i.e., occupation of one site influencing binding at another) that govern the concentration-dependence of ligand-induced behavior. Methods that inform on these processes are important for understanding mechanisms of ligand-activation and aiding the rational design of novel therapies targeting membrane receptors. However, structures and ensemble-averaged measures of distinct partially liganded intermediates are difficult to resolve due to averaging over transient asynchronous events and mixtures of heterogeneous bound states.

Single-molecule (SM) approaches are ideal for resolving both heterogeneous states and asynchronous dynamics. However, their application to studies of ligand binding in membrane proteins is challenged by costly sample preparation, dysfunction in non-native lipid or detergent environments[4–6], lack of solution access to intracellular sites, and nonspecific dye adsorption to imaging surfaces. Here, we overcome these challenges by combining cell-derived nanovesicles[7–9], microfluidics, micromirror total internal reflection fluorescence (mmTIRF)[10], and colocalization SM spectroscopy[11] to optically track binding and unbinding of individual fluorescently-tagged ligands at single membrane proteins. Imaging a ~100 μm × 100 μm field of view much larger than a confocal spot enables high-throughput data acquisition of up to hundreds of molecules simultaneously.

Cell-derived nanoscale vesicles are an attractive approach for isolating full-length membrane proteins in their native lipid environment and provide access to extracellular and intracellular sites due to the stochastic nature of vesicle formation. SM imaging of individual vesicles has been used to study nicotinic acetylcholine receptor (nAChR) stoichiometry[8] and mechanisms of membrane curvature sensing[12]. In contrast, live cell-based approaches offer a native membrane environment[13–15] but are challenged by cell autofluorescence, diffusion within the membrane, high local protein concentrations that limit SM resolution[16–18], and lack of access to intracellular domains. Isolating proteins with patch pipets allows simultaneous optical and electrical recording[19], but severely limits throughput to one molecule at a time. Alternative cell-free approaches such as solubilization in detergent or lipid synthetics can leave the protein locked into conformations that constrain normal function[4–6] or require stabilizing mutations that may impose unnatural constraints on protein structure and function[20,21].

As an exemplar system, we use our approach to resolve the binding dynamics of a fluorescent cyclic nucleotide analog (fcGMP[22,23]) to individual cyclic nucleotide-gated (CNG) TAX-4 ion channels from *C. elegans* in cell-derived nanovesicles. CNG channels are critical for visual and olfactory transduction[24]. They are tetramers comprised of four subunits surrounding a central ion-conducting pore[25]. Binding of cGMP or cAMP to intracellular cyclic nucleotide-binding domains (CNBDs), one per subunit, initiates the opening of the channel's cation conducting pore, thereby converting changes in cyclic nucleotide level to changes in membrane potential[24]. This allows photoreceptors in the visual system to respond to changes in light level via light-induced hydrolysis of cGMP, and olfactory receptor neurons to respond to the odorant-induced synthesis of cAMP[24,26]. Mutations in CNG channels have been linked to progressive vision loss and color vision abnormalities such as macular degeneration[27–31] as well as olfactory disorders such as isolated congenital anosmia[32]. Recently, a clinical trial of subretinal *CNGA3* gene therapy in individuals with complete achromatopsia resulted in significantly improved visual acuity and contrast sensitivity, thus validating CNG channels as a promising therapeutic target[33].

Cryo-EM structural models of TAX-4[34,35], human rod CNGA1 channels[36], and prokaryotic homologs[37,38] in both unliganded and cGMP/cAMP-bound conformations provide important static snapshots of endpoints during ligand activation. These structures as well as others from prokaryotic homologs[37,38] are complimented by functional studies of channel currents and measures of ligand-dependent conformational changes using approaches such as fluorescence spectroscopy[39], electron paramagnetic resonance[40], and high-speed atomic force microscopy[41]. However, these studies do not by themselves reveal the sequence of events connecting the observed structural endpoints, which requires temporal resolution of distinct bound states that have previously not been directly observed.

Our observations provide a first look at individual CNBD dynamics and initial binding cooperativity in a full-length CNG channel embedded in a native cell membrane. These results reveal similarity to individual site dynamics with structurally similar CNBDs from HCN channels and constrain plausible models of binding cooperativity. Our combined approach has broad application to other membrane receptors and complements structural information with dynamics for transient states whose structures are not easily resolved.

## Results

**Immobilization of cell-derived nanovesicles for single-molecule imaging.** For imaging and immobilization, we fused the enhanced green fluorescent protein (EGFP) to the cytosolic N-terminus of the CNG channel TAX-4 (GFP-TAX-4). Cell-derived nanovesicles containing GFP-TAX-4 were generated as described for a study of nicotinic acetylcholine receptor stoichiometry[7–9,42] (Fig. 1). Briefly, HEK-293T cells transfected with GFP-TAX-4 were disrupted using nitrogen cavitation, resulting in the spontaneous formation of cell membrane vesicles with diameters around 200 nm[9,42] (Supplementary Fig. 1). The vesicles were further separated by gradient ultracentrifugation into fractions comprised primarily of either plasma membrane (PM) or endoplasmic reticulum (ER) membrane (Supplementary Fig. 2). The PM vesicle fraction was applied to a microfluidic chamber on a passivated glass coverslip with a GFP-nanobody bait protein sparsely deposited on its surface for on-chip purification and immobilization of GFP-TAX-4[43,44]. Due to the stochastic nature of vesicle assembly, a mixture of vesicles with either extracellular or intracellular leaflets exposed to the bath solution is obtained[45]. Vesicles containing GFP-TAX-4 oriented with both GFP and the intracellular CNBDs outside the vesicle are immobilized upon binding to the GFP-nanobody on the chip surface (Fig. 1), whereas vesicles without GFP-TAX-4 or with GFP and CNBDs oriented toward the inside of the vesicle are washed away upon rinsing with buffer (Supplementary Fig. 3a). To verify the applicability of this technique to proteins with more complex stoichiometries and extracellular binding domains such as reported for nAChRs[8], we performed the same nanovesicle preparation and immobilization procedure with hetero-pentameric GABA$_A$ receptors comprised of $\alpha_1$, $\beta_2$, and $\gamma_{2L}$ subunits (Supplementary Fig. 3b).

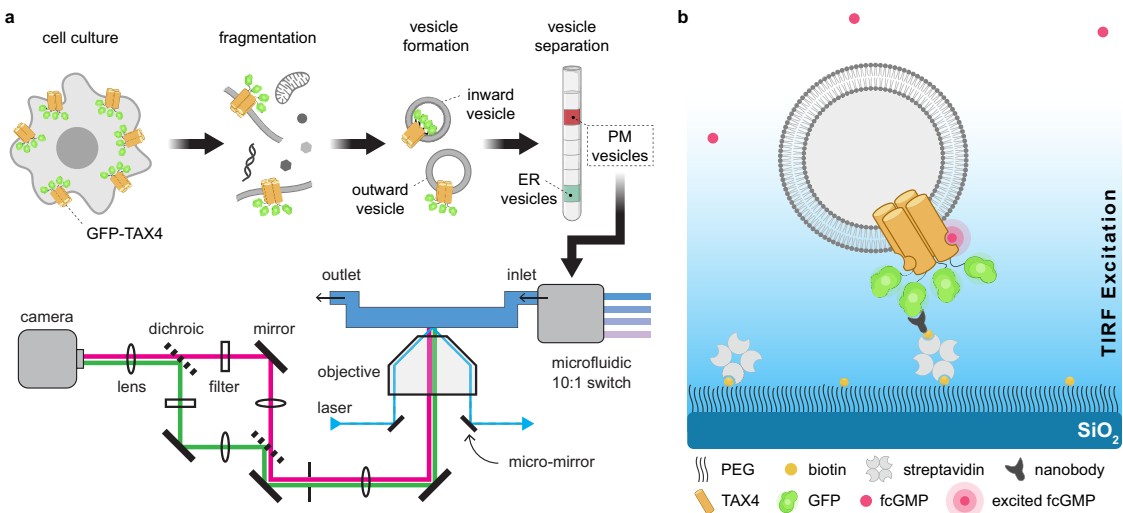

**Fig. 1 Immobilization of cell-derived nanovesicles for single-molecule imaging of membrane proteins. a** Schematic illustrating both sample preparation and the imaging setup. Briefly, cells expressing the membrane protein of interest (e.g., GFP-TAX-4) are fragmented using $N_2$ cavitation to spontaneously form nanoscale vesicles, some of which contain the protein of interest in a mixture of inward and outward-facing orientations. Vesicles comprised of membrane from either the plasma membrane (PM) or endoplasmic reticulum (ER) are further separated by gradient ultracentrifugation. The fraction of PM vesicles is applied to a sample chamber for immobilization and imaging with micromirror total internal reflection (mmTIRF) microscopy. **b** Illustration of an individual immobilized vesicle in the sample chamber. The chamber consists of a glass coverslip coated with a layer of PEG doped with PEG-biotin to which a biotinylated anti-GFP nanobody (bait protein) is attached via streptavidin. Vesicles containing GFP-TAX-4 oriented such that GFP is exposed to the extravesicular solution is immobilized at nanobody locations on the optical surface. TIRF excitation, indicated by the blue gradient, ensures that bulk freely diffusing fluorescent ligand (e.g., fcGMP) above the vesicle layer are not appreciably excited. Fluorescence from the vesicle layer is imaged on an EMCCD as depicted to the left. Note that the vesicle is not drawn to scale as it would on average have a diameter about 20-fold larger than GFP-TAX-4.

**Resolving individual binding events at TAX-4 CNG channels.** For optical detection of ligand binding events, the sample chamber was continuously perfused with a fluorescent cGMP conjugate (fcGMP) previously shown to activate CNG channels with similar efficacy and affinity to cGMP[22]. Importantly, TIRF limits excitation laser power to an evanescent field within ~100–200 nm of the optical surface, encompassing primarily the layer of immobilized vesicles only. This is critical, as illumination of the vast majority of freely diffusing fcGMP above the vesicle layer would result in background fluorescence precluding resolution of individual fcGMP molecules. Because the time for free fcGMP to diffuse through a diffraction-limited spot (~1 ms) is appreciably shorter than the time resolution of our recordings (50 ms per frame), we did not resolve diffusion of fcGMP to or from TAX-4, but only observed increased fluorescence when fcGMP remained at a spot for a time period comparable to or longer than the frame duration, as when bound to TAX-4.

Colocalized spots with both EGFP and fcGMP fluorescence were identified as immobilized vesicles containing GFP-TAX-4 with functional binding domains (Fig. 2). To determine whether colocalized binding events represent specific binding to TAX-4 rather than nonspecific adsorption to the surface or lipids, we tested the ability of non-fluorescent cGMP to outcompete fcGMP for its binding site. Using a microfluidic pump and switch we perfused a mixture of fcGMP and an excess of non-fluorescent cGMP to outcompete fcGMP binding at the same molecules previously imaged in fcGMP alone. Colocalization of EGFP and fcGMP fluorescence, consistent with specific binding to immobilized GFP-TAX-4 receptors, was largely abolished by competition with non-fluorescent cGMP and recovered upon reapplication of fcGMP alone (Fig. 2). This suggests that reversible binding at these locations reflects fcGMP specific association with TAX-4 CNBDs.

We also observed fcGMP fluorescence at spots devoid of EGFP signal. The similar frequency of these noncolocalized spots upon

competition with cGMP suggests that they reflect nonspecific adsorption of fcGMP (Fig. 2). To characterize noncolocalized signals, we compared the mean intensity and standard deviation of individual events to those at colocalized spots. Noncolocalized fcGMP events tend to have lower intensities and higher variance than colocalized events (Supplementary Fig. 4a). However, we occasionally observed noncolocalized high-intensity events that we interpret as adsorption of fcGMP to the coverslip where the TIRF excitation field is most intense (Supplementary Fig. 4b, c). In this case, the lower intensity noncolocalized events are likely to reflect adsorption to aggregates in the surface layer. The high-intensity events were generally much longer-lived than the vast majority of binding events, suggesting that termination of most colocalized binding events can be attributed to unbinding rather than fluorophore bleaching. Noncolocalized EGFP spots were also observed, which we hypothesize reflect vesicles where surface interactions occlude solution access to CNBDs or otherwise render them nonfunctional. To verify that colocalized fcGMP fluorescence signals were not attributable to nonspecific interactions with nanovesicle lipids, we performed the same fcGMP binding experiment with immobilized nanovesicles containing either a TRPV1 channel fused to an intracellular GFP or GABA$_A$ receptors that do not bind cGMP. We did not observe any appreciable colocalization in the absence of GFP-TAX-4 (Supplementary Figs. 5 and 6). Taken together, these observations strongly suggest that colocalized EGFP and fcGMP fluorescence spots reflect specific binding of individual fcGMP molecules to the CNBDs of immobilized GFP-TAX-4 channels.

To restrict our analysis to colocalized spots containing single CNG channels we estimated the number of GFP-TAX-4 subunits in each diffraction-limited spot by counting the number of bleach steps in the EGFP fluorescence time series (Fig. 3a). Given that TAX-4 channels are tetramers, we excluded from the analysis all spots with more than four bleach steps. The distribution of bleach steps for the remaining spots was well described by a binomial

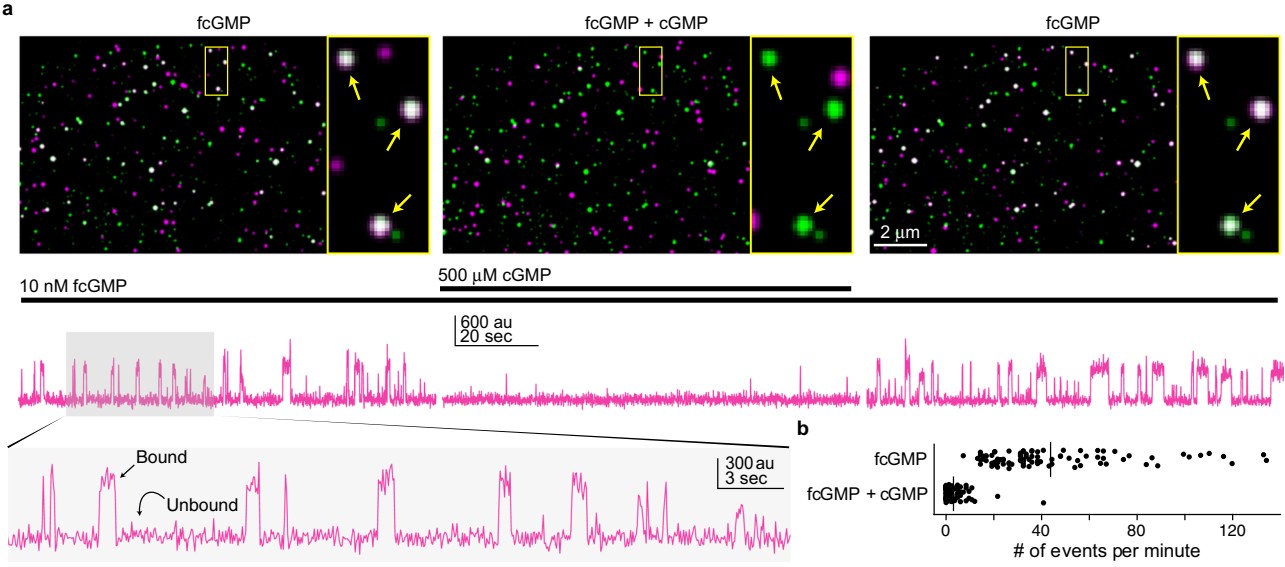

**Fig. 2 Single-molecule imaging of fcGMP binding to GFP-TAX-4 in cell-derived nanovesicles. a** Time-averaged fluorescence in arbitrary units (au) for GFP (green) overlaid with fcGMP (magenta). Colocalized GFP and fcGMP signals appear white. From left to right depicts sequential epochs for the same field of view showing binding in 10 nM fcGMP, block of specific binding by coapplication with an excess of 500 µM non-fluorescent cGMP, and recovery upon removal of cGMP. The yellow box is expanded to the right of each image and arrows indicate locations exhibiting specific fcGMP binding to GFP-TAX-4. Below each image is time series for fcGMP fluorescence at a single colocalized spot during each of the epochs in the corresponding images above. Transient increases in fluorescence reflect individual fcGMP binding events in the vesicle layer at the optical surface. The shaded region is expanded below. Results were similar for five separate experiments. **b** The frequency of binding events in 10 nM fcGMP is greatly diminished in the presence of 500 µM competing non-fluorescent cGMP. Source data are provided as a Source Data file.

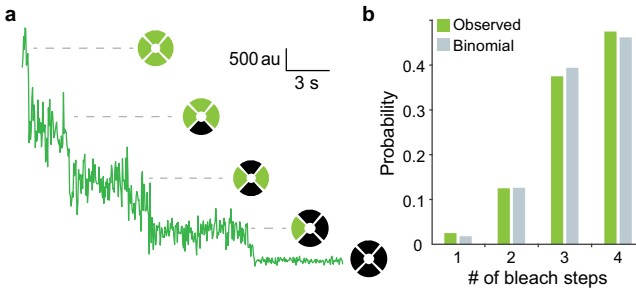

**Fig. 3 Number of GFP-TAX-4 subunits per vesicle assessed by photobleaching of GFP. a** Stepwise photobleaching of GFP fluorescence in arbitrary units (au) from a single diffraction-limited spot. **b** The number of observed bleach steps for colocalized spots included in the analysis. The good fit to a binomial distribution with four sites suggests that most spots contain a single tetrameric channel where bleaching of each GFP is observed with a probability of 0.82 and 95% CI [0.74, 0.89]. Source data are provided as a Source Data file.

distribution for four sites with an estimated probability of observing each EGFP bleaching event of 0.82, consistent with reports for this probability (Fig. 3b)[8,14]. These spots were considered as having a high probability of containing single tetrameric GFP-TAX-4 channels.

**TAX-4 CNBDs undergo a conformational change following binding**. We measured a total of ~60 h of SM binding dynamics across a range of fcGMP concentrations from 10 to 200 nM. Time series for fcGMP binding at these spots show a clear dependence on concentration as expected for binding events (Fig. 4a). At low concentrations of 10–30 nM fcGMP, we almost exclusively observed single isolated binding events. With increasing concentrations from 60 to 200 nM fcGMP, we observed an increasing frequency of simultaneous binding of multiple fcGMP at

individual molecules (i.e., stacked fluorescence steps). We did not explore higher fcGMP concentrations necessary to saturate the binding sites due to the challenge of increased background fluorescence from the freely diffusing ligand.

Fluorescence time series for fcGMP binding were idealized to obtain the number of bound fcGMP at each time point (Fig. 4b, Supplementary Figs. 7, 8, and see "Methods" section). The average bound probability across molecules exhibited a dependence on fcGMP concentration similar to observations of the cGMP concentration-dependence of channel current[46] (Fig. 4d), further supporting the idea that our observations reflect binding events associated with activation of TAX-4. Furthermore, bound lifetimes were independent of concentration, whereas unbound lifetimes monotonically decreased with increasing fcGMP concentration as qualitatively expected for a binding reaction (Fig. 5a, c). We assessed the ability of the idealization procedure to resolve individual events by applying it to simulated fluorescence binding data with lifetimes and noise drawn from the experimental observations (Supplementary Fig. 7). Comparison of the simulated and idealized event records indicates overall very good detection of singly and doubly liganded events except for brief single-frame events in doubly liganded states which were missed approximately one-third of the time (Supplementary Fig. 8; see "Methods" section).

Bound lifetimes at all tested fcGMP concentrations were poorly described by a single exponential distribution indicative of unbinding from a singular bound state. Instead, at least two exponential components were needed to account for our observations (Fig. 5b and Supplementary Fig. 15, Supplementary Table 1), suggesting that the CNBDs adopt multiple bound conformations. To determine whether these different conformations arise from two distinct populations of molecules or reflect conformational exchange within a single population of molecules, we examined correlations between the lifetimes of successive binding events at individual molecules (Fig. 5d). If shorter and

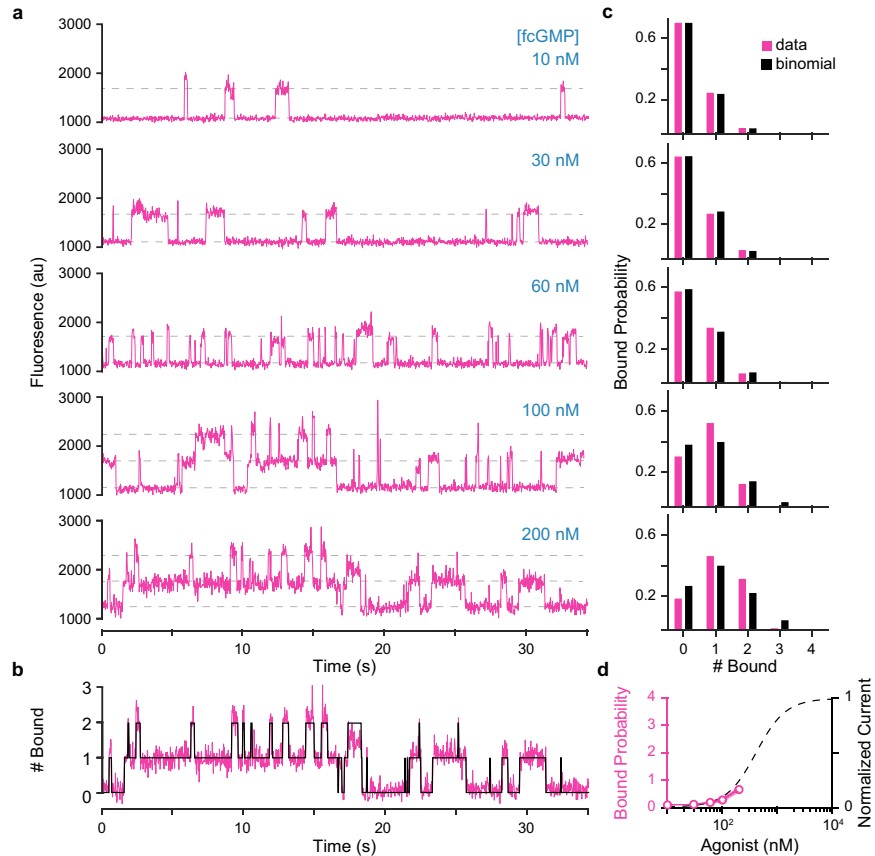

**Fig. 4 Concentration-dependent fcGMP binding. a** Time series for fcGMP binding to single TAX-4 containing nanovesicles at increasing concentrations of fcGMP. Horizontal dashed lines indicate approximate fluorescence levels reflecting stepwise binding of one or two fcGMP molecules. Fluorescence is in arbitrary units (au). **b** Idealization (black) of fluorescence (magenta) time series for the number of bound fcGMP at an individual molecule in 200 nM fcGMP (see "Methods" section). **c** Bound probability distributions for all tested concentrations and fits to a binomial distribution assuming four identical and independent sites. Bound probability [95% CI] per site from the binomial fits: 10 nM = 0.08 [0.079, 0.081], 30 nM = 0.10 [0.09, 0.11], 60 nM = 0.12 [0.10, 0.14], 100 nM = 0.21 [0.20, 0.22], 200 nM = 0.26 [0.24, 0.28]. **d** Average fcGMP bound probability across all molecules and normalized ionic current[46] as a function of fcGMP or cGMP concentration, respectively. Shaded area represents SEM (partially hidden by the line width). Current values are for fits of the Hill equation to inside-out patch clamp recordings in Komatsu et al.[46]. Source data are provided as a Source Data file.

longer bound lifetimes were attributable to distinct populations of molecules, we would expect little correlation between short and long bound events within individual molecules. In this case, we would expect to see primarily either short- or long-lived binding events at an individual molecule (i.e., lower left or upper right along the dashed diagonal line in Fig. 5d), and relatively fewer pairs of sequential short/long or long/short events (i.e., the off-diagonal in Fig. 5d). In contrast, we observe both short and long binding events at individual channels with high probability, suggesting that individual CNBDs exchange between at least two bound conformations. Unbound lifetime distributions were also better described by two exponentials, suggesting that unliganded CNBDs also adopt at least two conformations. However, one of the two exponential components accounted for the majority of the unbound lifetime distributions, consistent with the high-frequency occurrence of a single unliganded conformation.

To explore the dynamics of this process at individual CNBDs we evaluated a series of hidden Markov models (HMMs) ranging from the simplest possible two-state binding mechanism to models with up to two bound and unbound states (Supplementary Fig. 9). We restricted analysis of these models to isolated binding events by removing periods with two or more bound ligands, thereby splitting those time series into segments comprised only of singly-bound events. Models were globally optimized in QuB[47,48] for all molecules and concentrations and

ranked according to their relative Bayesian Information Criterion ($\Delta$BIC = BIC – $\text{BIC}_{\text{best model}}$) scores (smaller is better) (Supplementary Fig. 9, Supplementary Table 2, and see "Methods" section).

As expected, given the observed lifetime distributions, models with two bound states were preferred (smaller $\Delta$BIC) over that with a single bound state (Supplementary Fig. 9 and Supplementary Table 2). The addition of a second unbound state also slightly reduced $\Delta$BIC, although not significantly, suggesting that models with only a single unbound state are likely to be a reasonable approximation. Amongst models with one unbound and two bound states, we favor M1.B. Our rationale is that both linear models M1.B and M1.C are subsets of the cyclic model M1.D for which one of the pairs of optimized rate constants in the loop tended towards zero causing M1.D to essentially recapitulate M1.B. For models with two unbound and two bound states, the cyclic model M1.F suggests that binding/unbinding is approximately 10-fold less frequent following the conformational exchange, and thus we favor the simpler model M1.E which lacks these transitions. Both M1.B and M1.E suggest that binding is followed by a conformational change of the bound complex. Although less frequent, a conformational change of the unliganded site may be possible, in which case it appears to impair binding/unbinding. These results are consistent with the observations that both bound and unbound duration

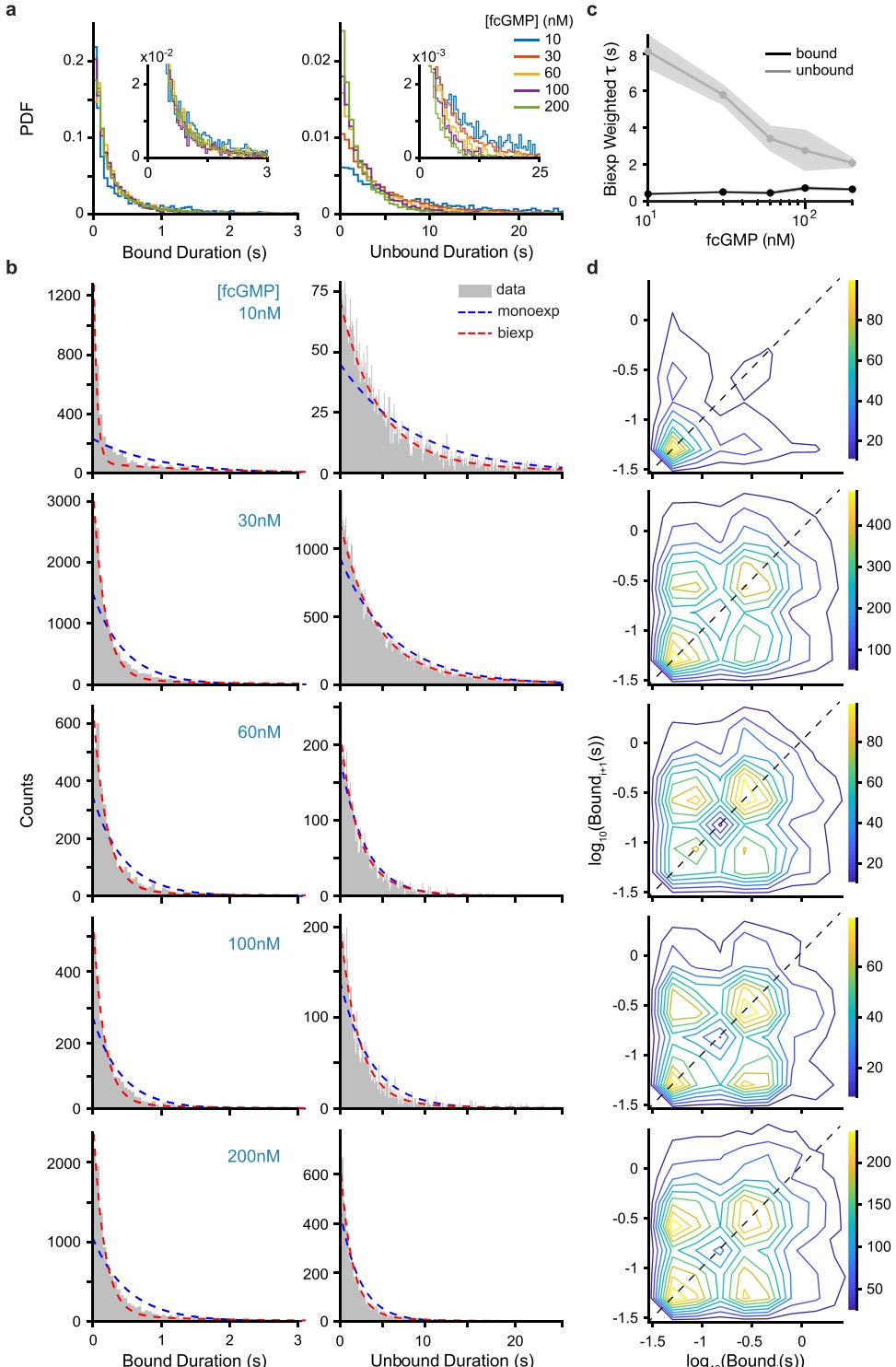

**Fig. 5 Bound and unbound dwell time distributions and correlations. a** Bound and unbound fcGMP dwell time distributions across all molecules from idealized records (see Methods). **b** Dwell time distributions for data (gray) overlaid with mono- (blue dashed) and biexponential (red dashed) maximum likelihood fits (Supplementary Table 1). **c** Weighted time constants from biexponential fits of bound (black circles) and unbound (gray circles) dwell times as a function of fcGMP concentration with associated 95% confidence intervals (shaded area). **d** Correlation between the duration of sequential singly-bound events $i$ and $i+1$ within individual molecules. The concentration of fcGMP is the same as for the dwell time distributions to the left. Color bar denotes the number of events. If short and long events arise from distinct populations of molecules, we would expect to observe clusters of events primarily along the dashed diagonal, whereas events on the off-diagonal represent sequential short and long bound durations within individual molecules. Source data are provided as a Source Data file.

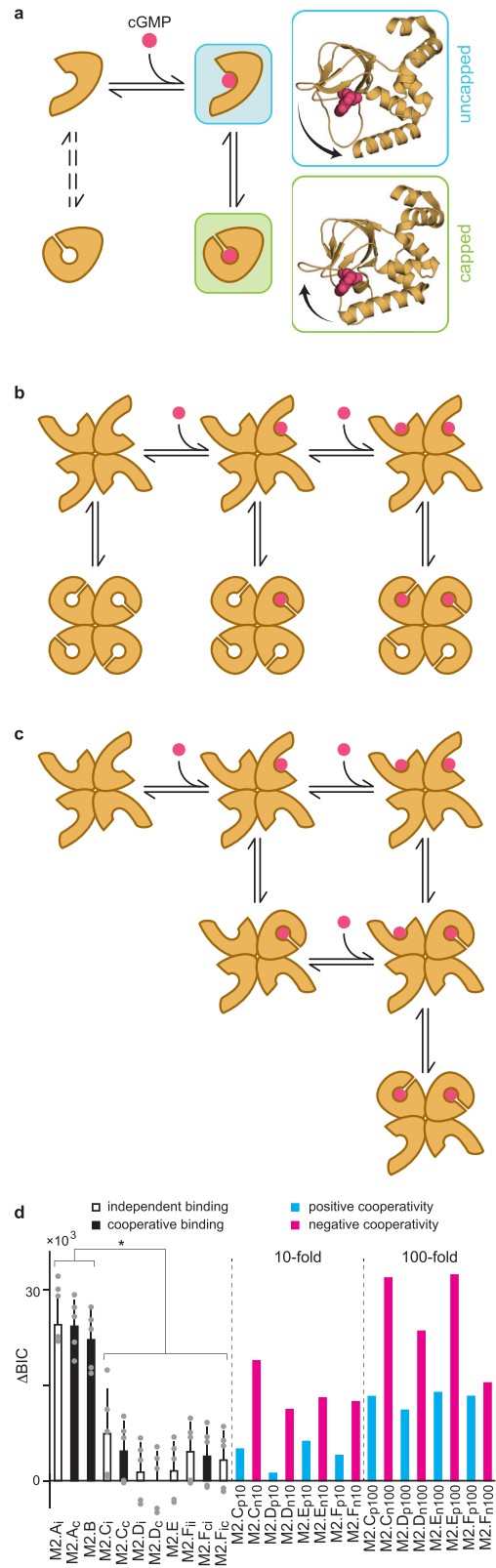

**Fig. 6 Models for the first and second binding steps. a** The preferred model for binding dynamics at individual CNBDs. This model depicts both a ligand association step (horizontal transition) as well as a conformational change of the CNBD in both unliganded and liganded states (vertical transitions). The dashed arrows indicate transitions that have a relatively smaller contribution to the observed dynamics. Postulated structures for the ligand-bound conformations are shown to the right. They are based on cryo-EM structures of TAX-4 in cGMP-bound (green) or unliganded (blue) conformations[34,35], where cGMP has been added to the unliganded structure in the position that it is found in the bound structure, and arrows denote the observed movement of the C helix between the two structures. Capping of the binding site by the C helix limits binding and unbinding, similar to dynamics observed at isolated CNBDs from HCN2 channels[49]. See Supplementary Fig. 9 and Supplementary Table 2 for all explored models of individual CNBDs and their optimized rate constants. **b, c** Two potential models depicting binding of the first two ligands and either a global conformational change of all four CNBDs in each ligation state (b) or individual conformational changes of each ligand-bound CNBD (**c**). We were unable to distinguish between these mechanisms. See Supplementary Fig. 10 and Supplementary Table 3 for all explored models of the first two binding steps and their optimized rate constants. **d** BIC scores for all tested two-site models relative to the model with the best score (smaller score is better; $\Delta BIC = BIC - BIC_{\text{best model}}$). See Supplementary Fig. 10 and Supplementary Tables 3–5 for model descriptions and their optimized rate constants and constraints. Bars are $\Delta BIC$ scores for model fits to the entire data set. Error bars are standard deviations across BIC scores for five randomized folds of the data set (data points are BIC scores for each fold mean-centered on their associated bars). Significance was determined by one-way ANOVA with post-hoc Tukey test at *$p < 0.05$. See source data file for exact $p$-values. See Supplementary Table 5 for details on constraints for models with defined 10- or 100-fold positive or negative cooperativity. Source data are provided as a Source Data file.

capping of the binding pocket by the C helix[49] may also occur for TAX-4 CNBDs (Fig. 6a).

**The first two binding steps.** A fundamental limitation to fluorescence imaging is that background fluorescence and noise increase with the increasing concentration of fluorescent dye. At micromolar concentrations of fcGMP, this background noise challenges the identification of individual binding events. Thus, we limited our observations to concentrations that did not saturate the four binding sites in individual channels. Nonetheless, we were still able to resolve simultaneous ligand binding events as distinguished by multiple stepwise increases in fluorescence. At the highest concentration tested (200 nM) most events (99%) reflect 0, 1, or 2 bound ligands. We, therefore, restricted our analysis to only the first two binding steps by removing periods with more than two bound ligands from the binding time series, thereby splitting those records into segments. We explored the ability of several two-site HMM models to explain the observed binding dynamics across all molecules and concentrations and ranked them according to their relative $\Delta BIC$ scores (Supplementary Fig. 10, Supplementary Table 3, and see "Methods" section). Note that subscripts on the model labels indicate different sets of constraints as described in Supplementary Fig. 10 and Supplementary Table 3.

The simplest model for sequential binding to two identical and independent sites (M2.A$_i$) has the worst ranking ($\Delta BIC$), which is hardly improved by allowing nonidentical (i.e., cooperative) binding between the two sites (M2.A$_c$). Strikingly, an extension of M2.A$_c$ with two distinct doubly bound states (e.g., adjacent versus diagonal subunits) (M2.B) also provides very little improvement in $\Delta BIC$. In contrast, all additional models tested that include

distributions are biexponential, but that one of the components dominates for the unbound distributions. Model M1.E (Fig. 6a and Supplementary Fig. 9) has the same form as the model we previously used to describe dynamics at individual isolated CNBDs from HCN2 channels[49]. Thus, we hypothesize that a similar mechanism to that proposed for HCN2 CNBDs involving

both binding and a conformational exchange of the bound complex (M2.C-F) were similarly preferred over models M2.A-B (Supplementary Fig. 10 and Supplementary Table 3). Notably, model M2.$F_{ii}$ has the same number of free parameters (four) as M2.$A_c$, but is much preferred. Similarly, M2.$C_i$, M2.$F_{ci}$, and M2.$F_{ic}$ have the same number of free parameters (six) as M2.B, but are much preferred. Interestingly, allowing nonidentical (i.e., cooperative) binding sites uniformly provided only a slight nonsignificant improvement in ΔBIC as compared to models constrained to identical and independent sites. Where allowed, the second binding/unbinding step was typically a few folds faster/slower than predicted for independent sites. Although larger number of free parameters (8-10) in cooperative models M2.$C_c$ and M2.Dc raises some uncertainties regarding their uniqueness, qualitatively similar results were observed for models M2.$F_{ci}$ and M2.$F_{ic}$ with only six free parameters each. Furthermore, rate constants for models M2.$C_c$ and M2.Dc are similar to those for our preferred single-site model M1.E where a comparison is valid. Finally, simulated bound and unbound dwell-time distributions for models M2.D-F provide a reasonably good description of our observed dwell-time distributions, whereas those for M2.A-B are noticeably worse, recapitulating the ΔBIC ranking (Supplementary Fig. 11). We note that to keep the models tractable, we did not consider the combination of distinct doubly bound orientations (e.g., M2.B) and conformational changes separate from ligand association/dissociation. Instead, we focused on the conformational change whose importance is suggested by both single-site dynamics and the preference for models M2.C-F over M2.A-B.

Based on ΔBIC we cannot distinguish between models M2.C-F. Nonetheless, we favor either model M2.D or M2.F that describe sequential binding to two sites and a conformational change that occurs either globally in both sites together (Fig. 6b; M2.D) or independently at each site (Fig. 6c; M2.F). Our rationale is as follows: Each of models M2.$C_i$, M2.E, M2.$F_{ii}$, M2.$F_{ci}$ and M2.$F_{ic}$ have 4-6 free parameters and are significantly preferred as compared to models M2.$A_c$ and M2.B also with four or six free parameters, respectively. This suggests that the addition of a conformational exchange in distinct ligation states is a primary factor for the improved ΔBIC, and that our data can distinguish between consistent and inconsistent models with these number of free parameters. Although models M2.$C_c$ and M2.$D_i$ have eight free parameters, their optimized rate constants were very similar to those of M2.$C_i$ with less than 2-fold changes in binding/unbinding rates. Thus, the additional two free parameters as compared to M2.$C_i$ did not provide sufficient flexibility to find a completely different set of rates. Similarly, model M2.$D_c$ with 10 free parameters was very similar to M2.$C_c$. The similarity of models M2.C and M2.D which differ only in the inclusion of an unliganded conformational exchange is also consistent with observations from single-site models where exclusion of this less frequently visited state did not have a major impact on ΔBIC. Thus, we still like models M2.D (or the subset M2.C) despite their larger number of free parameters, although we acknowledge that the extra free parameters raise a bit of ambiguity for these models.

Model M2.E describes a Monod–Wyman–Changeux (MWC) model where ligand affinity depends on the global conformation and the equilibrium between conformations is influenced by the number of bound ligands. Although optimized parameters for M2.E imply that binding of each ligand increases the equilibrium constant for the conformational exchange by a factor of 1.6, they also indicate that the dynamics of the exchange are slowed upon ligand binding by approximately 5- to 10-fold. Given the constraints in the model, we suspect that the slower exchange may be an artifact of a requirement to describe slower association/dissociation after the conformational change as

observed for dynamics at single sites (M1.F). Also, the optimized conformational exchange rates for M2.E are much slower than for any of the other models including those for single sites. Thus, we prefer M2.D (or its approximation M2.C) where presumably less frequent binding/unbinding following the conformational change is ignored, or alternatively M2.F where the conformational exchange occurs separately for individual subunits.

Together, these results suggest that (1) adjacent versus diagonal patterns of site occupation in the channel tetramer are insufficient by themselves to describe dynamics for the first two binding events and (2) multiple bound conformations for distinct ligation states is an important feature of cGMP association with TAX-4, consistent with our observations for dynamics at single sites. However, we cannot distinguish between a global (Fig. 6b; M2.D) or per-subunit (Fig. 6c; M2.F) conformational change. Additional binding data for the third and fourth steps will likely be needed to distinguish these possibilities.

**Binding cooperativity.** We take binding cooperativity to reflect the degree to which durations in doubly bound states differ from that expected for two independent and identical sites. Our single-molecule binding observations provide several ways to explore cooperative binding between the first two ligands. First, since we resolve the time spent with distinct numbers of bound ligands, and hence their probabilities, we can compare them to a binomial distribution as expected for identical and independent sites. Across tested concentrations, bound probability distributions were fairly well described by a binomial distribution with the probability of each site being occupied increasing with ligand concentration (Fig. 4c). Thus, approximating the first two binding steps as identical and independent provides at least a rough approximation of this data. However, at higher concentrations, the binomial fits tend to slightly overestimate the probability for no ligands to be bound and underestimate the probability for one and/or two ligands to be bound, which could reflect positively cooperative interactions between the first two ligands (Fig. 4c).

Second, we compared our experimental data to simulations of independent binding sites (see "Methods" section). For independent sites, the distribution of durations in doubly bound states should be truncated as compared to the distribution of durations in singly-bound states. This is because statistically, the rate of unbinding at either of two identical and independent sites should be twice that of a single site. As expected, simulations of independent sites predict fewer longer-lived periods with two bound ligands as opposed to one bound ligand, which we can resolve with our idealization procedure after adding noise analogous to our experimental recordings (Supplementary Fig. 12a, b). In contrast, we do not observe a similar reduction in the frequency of longer-lived doubly bound events in the experimental fcGMP binding series (Supplementary Fig. 12c). It is difficult to rule out that such an effect might be present, but it appears to be at least reduced as compared to the simulated prediction for independent sites. This suggests that longer-lived doubly bound states are further stabilized as compared to their prediction for identical and independent sites, although the effect is somewhat subtle. We note, however, that we could be underestimating this effect given that the longest-lived events may be truncated by bleaching (Supplementary Fig. 14).

Third, we compared the latency to binding of the second ligand (2nd latency) after binding the first ligand in our experimental observations with predictions from simulations of independent sites. The distribution of 2nd latencies was shifted to shorter durations in the experimental fcGMP bound series as compared to simulations of independent sites, again suggesting that the binding rate for the second ligand is slightly faster than that of the

first (Supplementary Fig. 13). The ratio for the mean 2nd latency of fcGMP binding data and independent simulations suggests that the second ligand binds approximately 1.3-fold faster than predicted for independent sites, similar to predictions in tested models.

Fourth, we compared models where the first and second binding steps were constrained to be identical and independent (M2.C$_i$, M2.D$_i$, M2.F$_{ii}$, M2.F$_{ic}$) or allowed to be cooperative (M2.C$_c$, M2.D$_c$, M2.F$_{ci}$) (see constraints in Supplementary Table 3). Removing the constraint for independent association at each site led to models predicting an increase in the binding equilibrium of the second ligand (i.e., positive binding cooperativity) (Supplementary Tables 3 and 4). However, the predicted increases are not large, with factors in the range 2.1–5.8. Furthermore, cooperative models were not distinguishable from those with identical and independent binding sites based on ΔBIC scores (Fig. 6d).

To further evaluate the ability of our data set to distinguish between various types of binding cooperativity we constrained the equilibrium constant for the second binding step in models M2.C, M2.D, and M2.F to be either 10- or 100-fold larger (positive cooperativity) or smaller (negative cooperativity) than expected for identical and independent sites (Supplementary Table 5). For simplicity, we split the cooperative factor evenly between binding and unbinding rates. For model M2.E, we similarly constrained the factor by which ligand binding influences the conformational exchange equilibrium. Overall, models with 10-fold positive cooperativity have similar ΔBIC scores to the most likely two-site models, whereas models with 100-fold positive cooperativity or 10- or 100-fold negative cooperativity are much less preferred (Fig. 6d).

Taken together, these results suggest that either the first two binding steps are identical and independent or that cooperative interactions between the sites confer up to a ~10-fold increase in the binding equilibrium constant for the second ligand.

## Discussion

Although multicolor SM fluorescence has previously been used to resolve binding dynamics for soluble proteins or nucleotides[50], similar measures for small molecule binding to membrane proteins in native lipids are few. For oligomeric proteins, the ability to resolve dynamics in distinct ligation states (i.e., singly versus doubly bound) is crucial for exploring the sequence of binding events and their cooperative interactions. Here, we introduce a combined approach using cell-derived nanovesicles, microfluidics, and mmTIRF colocalization SM fluorescence spectroscopy to study single-receptor ligand binding in native lipids. Our approach has several advantages for SM measurements in membrane proteins: (1) Proteins are never removed from their cellular lipid environment. (2) Vesicles can contain the protein in both inward and outward-facing orientations, providing access to either extracellular or intracellular receptor sites. (3) Microfluidic liquid handling enables within-experiment solution exchange to readily identify molecules exhibiting specific binding amidst background signals from nonspecific adsorption that challenge typical SM colocalization experiments at the dye concentrations used. (4) Expression and on-chip purification of full-length proteins for SM experiments require only standard transfection of cultured cells as opposed to potentially costly and time-consuming searches for appropriate purification conditions in synthetic environments and stabilizing mutations. (5) Immobilization of proteins simplifies the observation of individual molecules over longer time periods to resolve dynamics. As an exemplar system, we applied this approach to study the first two cyclic nucleotide-binding steps to full-length CNG TAX-4

channels. Our results suggest that binding is followed by a conformational change of the bound complex similar to that observed in isolated CNBDs from HCN2 channels. Furthermore, our observations provide an unprecedented view of the dynamics of the first two binding events which place constraints on the degree of binding cooperativity between the first and second ligand.

The observed binding dynamics at individual CNBDs in TAX-4 are similar to observations of fcAMP binding dynamics at isolated CNBDs from HCN2 channels[49]. Thus, it is likely that similar intrinsic fluctuations of isolated CNBDs occur in the channel complex. Furthermore, a recent study arrived at an analogous conclusion that a conformational change follows binding in full-length HCN1 and HCN2 channels[51]. Based on both fcAMP dynamics and structures of apo and holo CNBDs, we previously postulated a mechanism for HCN2 whereby the conformational change involves capping of the binding pocket by the C helix, thereby hindering unbinding prior to uncapping[49] (Fig. 6a). This mechanism is also consistent with Förster resonance energy transfer (FRET)[52] and double electron-electron resonance (DEER)[53] measurements, and it provides a physical barrier that explains the lower frequency of binding/unbinding predicted to occur following the conformational change (e.g., compare models M1.E and M1.F in Supplementary Fig. 9). Although the data presented here does not provide direct evidence for this mechanism, it is consistent with a similar C helix motion occurring in TAX-4.

Cryo-EM structures of TAX-4 have recently been resolved in both unliganded and fully cGMP-bound states[34–36]. These structural snapshots show that the C helix moves towards the binding pocket when the ligand is bound. This motion as well as a rotation of the CNBDs about the pore axis and translation towards the lipid membrane have also been inferred for CNG channels with fluorescence[39], DEER[40], and high-speed atomic force microscopy[41]. However, the structural consequences of partially occupied CNBDs remain to be determined.

The most likely models explored here predict that the probability of the CNBDs to adopt their alternate conformation increases with successive binding of each of the first two ligands, suggesting that the conformational change is part of the ligand-activation pathway. However, it is important to keep in mind that our observations do not include channel current, which challenges direct comparison with previous models of CNG channel pore gating. Nonetheless, the dynamics of the conformational change at the CNBDs is too slow to account for single-channel observations of pore gating[54], and thus we hypothesize that it reflects an earlier step in the activation process preceding the opening of the pore gate. As such, changes in CNBD conformation would place the channel in a preactivated state from which bursts of channel openings could occur. CNBDs in mixtures of bound conformations could also underlie observed subconductances in channels covalently locked into distinct ligand-bound states[55].

Although our observations of binding dynamics strongly support a conformational change in ligand-bound CNBDs, it is unclear whether this is a global change involving all subunits or independent exchange within individual CNBDs. Of interest is a model that extends a simple MWC mechanism for ligand binding and pore opening[56] to postulate that the tetrameric channel behaves as a dimer of dimers[57]. However, given that we only observed the first two binding events, we cannot distinguish between channels functioning as tetramers or coupled dimers.

Combined simultaneous observations of fcGMP binding and channel current in macroscopic ensembles of CNGA2 channels previously led to a model with a structure similar to that shown in Fig. 6b, but where the transitions we associate with CNBD

conformational exchange were associated with pore gating[22]. This model predicts strong negative cooperativity exclusively during binding of the second ligand as compared to the first, and similarly strong positive cooperativity upon binding the third ligand (which is similar to the first), with a reduction/increase in binding equilibrium constants of over three orders of magnitude for the second/third binding steps, respectively. Our results suggest that such strong cooperativity for the second binding step does not occur in TAX-4, and furthermore that the second binding step may involve positive rather than negative cooperativity. Either CNGA2 channels behave differently than TAX-4, or models with more modest cooperative effects[58] are more likely. We hypothesize that the occlusion of individual binding events in distinct ligation states by ensemble averaging challenged the unique identification of cooperative effects in the prior study. Ultimately, additional experiments are needed to resolve the dynamics of the third and fourth binding steps, possibly with FRET between fcGMP and an acceptor label on the channel to retain SM resolution at higher fluorophore concentrations[59]. A recent study using similar single-molecule methods shows that cAMP binding is non-cooperative in closed HCN1 and HCN2 channels[51]. However, whereas CNG channels are directly gated by cGMP, HCN channels are gated primarily by voltage and modulated by cAMP. Thus, potential positive cooperativity for the second binding step may reflect ligand-induced pore opening in TAX-4, although this remains to be determined.

This study provides unprecedented observations of the dynamics of early binding events and a conformational change that precedes channel opening. Our observations complement static structural snapshots of fully unliganded and fully bound conformations by reporting on the dynamics of partially liganded conformations that lie on the reaction pathway between endpoints designated by recent cryo-EM structures. These data also exemplify how binding dynamics can be used to observe conformational exchanges that may otherwise be difficult to measure, either due to their transient nature, small structural motions, or because it is unclear where to place probes for more targeted measurements. Finally, our combined SM approach has broad application to any membrane protein where protein and/or ligand are amenable to fluorescent labeling.

## Methods

**Constructs**. TAX-4 was a gift from Drs. Jonathan Pierce and Iku Mori. EGFP was appended to the N-terminus of TAX-4 (GFP-TAX-4) in the pUNIV vector as follows: A previous construct with an N-terminal EGFP in pUNIV containing an ApaI restriction site between EGFP and the gene and an MluI restriction site following the gene was used as a starting template. The previous gene was excised between ApaI and MluI sites, and TAX-4 was inserted using the same sites. This resulted in a two-residue linker (G-P) between EGFP and TAX-4. The entire gene was sequenced for verification.

The full-length rat GABA$_A$ receptor α1, β2, and γ2L subunits in the pUNIV vector were a gift from Dr. Cynthia Czajkowski. mScarlet (Addgene #99280) was inserted in the M3-M4 loop of the α1 subunit between residues V372 and K373 using an in-frame non-native Asc1 restriction site (GGGCGCGCC) introduced through site-directed mutagenesis as previously described[60]. This results in the insertion of an additional three residues (G-R-A) on each end of mScarlet. EGFP was similarly inserted in the N terminal region of the β2 subunit between residues N4 and D5, again using an in-frame non-native Asc1 restriction site as described for mScarlet.

**Preparation of cell-derived nanovesicles**. HEK-293T cells were cultured at 37 °C and 5% CO$_2$ (Eppendorf). Cells were plated in 60 mm dishes and transfected with 1 μg of GFP-TAX4 and 3 μg of PEI-MAX per dish. After 24 h, cells from four dishes were combined and vesicles were prepared as previously described[9]. Briefly, the cells were subject to nitrogen cavitation at 600 psi. for 20 min while suspended in 3 ml of hypotonic protease inhibitor solution (in mM: 10 Tris-HCl, 10 NaCl, 1.5 MgCl$_2$, 0.2 CaCl$_2$, pH 7.4). One Pierce protease inhibitor tablet (ThermoScientific) was added per 10 ml of buffer. To separate plasma membrane vesicles from organelle membrane vesicles, the lysate was dispensed onto a gradient containing 60, 30, 20, and 10% solutions of OptiPrep, followed by ultracentrifugation at 112,000 × g for 90 min at 4 °C. Following centrifugation, the sample was

fractionated into nine 1–1.5 ml fractions using a peristaltic pump, where the highest density fraction was collected first. OptiPrep was removed from fractions containing plasma membrane nanovesicles via centrifugation at 100,000 × g for 1 h at 4 °C using a fixed angle rotor. The resulting pellet was resuspended in 250 μl of buffer for immobilization as described below.

**Imaging fcGMP binding at TAX-4 channels in cell-derived nanovesicles**. Prior to immobilization, coverslip glass was UV-cleaned (Jelight) and passivated with a PEG monolayer sparsely doped with PEG-biotin (Laysan Bio) as previously described[44,61]. A 50 μl microfluidic chamber (Grace Biolabs) was adhered to the passivated coverslip to allow for solution flow during imaging. The chamber was serially incubated in 10 mg/ml bovine serum albumin (BSA), 50 μg/ml streptavidin, and 1 μg/ml biotinylated GFP-nanobody (ChromoTek, gtb-250). All solutions were made in a buffer that consisted of phosphate-buffered saline (PBS; pH 7.4) supplemented with 1 mM CaCl$_2$ and 1 mM MgCl$_2$. Following each 10–15 min incubation step, the chamber was washed with 4 ml of buffer to remove any non-immobilized components. Finally, the chamber was incubated with a preparation of nanovesicles from cells expressing GFP-TAX-4 (i.e., protein of interest fused with GFP) for 10–15 min followed by washing with 4 ml of buffer. This resulted in sparse immobilization of vesicles containing GFP-TAX-4. The chamber was then connected to a microfluidic pump and switch (Elveflow) for perfusion and exchange of solutions containing various concentrations of 8-(2-[DY-547]-aminoethylthio) guanosine-3′,5′-cyclic monophosphate (fcGMP; BioLog) and cGMP in buffer. Solutions were continuously perfused at a constant flow rate of approximately one chamber volume per minute during imaging on an inverted mmTIRF microscope (Mad City Labs) under either 488 or 532 nm laser excitation (Coherent OBIS). Laser power at the sample was 20 W/cm$^2$ (488 nm) and 40 W/cm$^2$ (532 nm). Fluorescence emission from immobilized vesicles in response to mmTIRF excitation was recorded simultaneously for an ~100 × 100 μm field of view on a 512 × 512 EMCCD camera (Andor iXon) at a frame rate of 20 Hz (50 ms per frame) using Micro Manager 1.4 for image acquisition. For each field of view, we initially determined the location of all immobilized vesicles by bleaching the EGFP emission from 500 to 550 nm under 488 nm excitation. Thereafter, the binding of fcGMP in solution was monitored by recording emission from fcGMP within 560–950 nm under 532 nm excitation.

**Single-molecule fcGMP binding image analysis**. Time-averaged fluorescence for EGFP and fcGMP from the same field of view were overlaid to identify diffraction-limited spots containing GFP-TAX-4 that colocalized with fcGMP binding. For EGFP this average included only the first 20-50 frames prior to significant bleaching of the EGFP signal. The entire time series was averaged for fcGMP as unbleached fcGMP from the bath can continuously diffuse into the vesicle layer where binding is detected. Mechanical drift of the stage parallel to the imaging plane was subpixel during imaging even for tens of minutes or longer due to the high stability of the Mad City Labs mmTIRF stage. A small offset of up to a few pixels was sometimes observed between EGFP and fcGMP image sets due to mechanical perturbation from manual swapping of filters on the optical table. This offset was corrected by registering images of the time-averaged fluorescence for EGFP and fcGMP recordings in MATLAB using an affine transform. Drift perpendicular to the image plane was continuously autocorrected during imaging by a nano-positioning stage that adjusted the distance between the sample chamber and microscope objective to maintain the position of the mmTIRF excitation laser on a quadrant position detector downstream of the exit micromirror (Mad City Labs). For colocalized spots with both EGFP and fcGMP fluorescence, the average intensity within a five-pixel diameter circle centered on the spot was projected across each frame to obtain the fluorescence time series at that spot. Bleach steps for EGFP at each and every colocalized spot were manually evaluated to estimate the number of GFP-TAX-4 subunits in each spot and spots with more than four EGFP bleach steps consistent with multiple channels were excluded from the analysis. The number of molecules at each concentration was: 10 nM: 112, 30 nM: 325, 60 nM: 63, 100 nM: 65 and 200 nM: 132. In total, we recorded ~60 h of single-molecule binding.

To assess the impact of photobleaching on the observed event durations we examined the lifetimes of noncolocalized fcGMP events that we assume largely reflect adsorption to the surface and subsequent bleaching. If some of these events are truncated by unbinding from the surface rather than bleaching, we will underestimate the actual bleach times. Given that these events were typically noisy, we manually estimated their lifetimes for a handful of randomly selected experiments. The mean bleach time across molecules was 23.4 s, an order of magnitude longer than the time constant for the longest duration bound component (Supplementary Fig. 14 and Supplementary Table 1). This suggests that most binding events were terminated by unbinding rather than bleaching, although the lifetimes of the longest-lived bound events are likely to be slightly underestimated due to bleaching.

**Idealization of time series for number of fcGMP bound**. To extract time series for the number of bound fcGMP we first denoised fcGMP fluorescence time series using DISC[62]. A complication in interpreting these time series is that the fluorescence intensity for individual events was somewhat variable, challenging

assignment of the number of bound ligands at each time point based solely on intensity. The distribution of individual event intensities is close to normally distributed with a slight skew toward higher intensities (Supplementary Fig. 7), suggesting that this variation reflects a random process rather than distinct numbers of bound ligands or discrete subpopulations of receptors. Such variation is typically observed in other colocalization fluorescence experiments. The source of this heterogeneity is uncertain but is likely caused by shifts of the molecule in the exponentially decaying excitation field or dye photodynamics[63,64]. To address overfitting due to fluctuation of individual event intensities, neighboring piecewise constant segments identified by DISC were recursively merged if the difference between their mean fluorescence was less than the weighted sum of the standard deviations of their mean fluorescence, where the weight for each segment was the ratio of the number of data points in the segment to the total number of data points in both segments. Single-frame segments with intensities intermediate to their surrounding segments were visually identified as typically reflecting noise rather than discrete transition intermediates, and thus were merged into the neighboring segment with the most similar mean intensity. The denoised fluorescence time series was then constructed from the mean fluorescence within each discrete segment. The lowest intensity level identified by DISC was assumed to reflect the unliganded baseline, as even at the highest concentration tested periods of low fluorescence consistent with background were observed for all molecules. Any denoised segment containing any of these baseline data points was assigned a bound count of zero. For each contiguous block of remaining segments, the number of bound fcGMP was incremented or decremented based on whether the denoised intensity increased or decreased. Binding/unbinding events whose denoised intensity change was more than double that of neighboring unbinding or binding events, respectively, were considered to involve the binding or unbinding of two molecules of fcGMP within a single frame. Similarly, contiguous triplets of events that returned to within 100 au of their initial intensity were considered to include a single event for association or dissociation of two fcGMP within a single frame. In a few cases, this procedure resulted in a bound count of less than one for a segment we previously identified as having a nonzero number of bound ligands, suggesting that we misidentified the number of fcGMP associated with a prior individual event. To correct this, we recursively examined the chain of monotonic unbinding events preceding each segment with an erroneous bound count of zero. If this chain contained a segment with only a single frame, the two surrounding unbinding events were merged into a single unbinding event, otherwise, the number of fcGMP associating during the binding event preceding this chain was incremented by one. This resulted in an idealized series for the number of bound fcGMP from 0 to 4 at each time point.

To assess the ability of the idealization procedure to resolve individual events, we applied it to simulated binding data at 30 or 200 nM ligand concentration whose lifetimes and noise were drawn from the experimental fluorescence observations (see below and Supplementary Fig. 7). Comparison of the known and idealized event records for simulated binding data indicates that the idealization procedure is incredibly accurate at identifying singly and doubly liganded events lasting two or more frames (≥100 ms) but misses 36% brief single-frame dwells in doubly liganded states (Supplementary Fig. 8). This is due to the additive noise from multiple fluorescent ligands coupled with a dwell time on the order of our sample duration and is much less of an issue for single-frame events with only one occupied site (10% missed). Thus, brief single-frame intensity fluctuations following binding of one or more ligands that were identified as noise during idealization (e.g., see couple of spikes around 15 s in Fig. 4b) could include missed brief events at higher ligation states. The algorithms overall accuracy, precision and recall was computed allowing for misidentification of the exact timing of individual events by up to four frames (accuracy = 0.83, precision = 0.95, recall = 0.87, F1-score = 0.91).

**Dwell time distributions**. Bound and unbound dwell time distributions were obtained from the idealized time series. The primary purpose of these distributions is to inform on plausible kinetic models for the dynamics. For example, the biexponential bound duration distributions suggest at least two distinct bound states. However, for bound periods with the simultaneous occupation of multiple sites, it is ambiguous as to the order in which the ligands unbind. Thus we estimated the bound lifetime at each site by randomly assigning each unbinding event to one of the bound molecules (Fig. 5a, b). Repeating this random assignment led to nearly identical distributions, indicating that randomization itself did not severely distort the distributions. Given that our data do not unambiguously determine the order of unbinding, we further computed bound dwell time distributions under the following two extreme assumptions: (1) the first ligand to bind is always the first to unbind, and (2) the first ligand to bind is always the last to unbind (Supplementary Fig. 15). In all cases, bound duration distributions required at least two exponential components to describe the data, consistent with models containing two distinct bound conformations.

**Simulated fcGMP binding time series**. To simulate binding data resembling our experimental recordings, we used the idealized bound time series (see above) to extract baseline Gaussian noise and distributions of bound and unbound dwell times as well as isolated binding event intensities and their Gaussian fluctuations. To address periods with multiple bound ligands, we estimated individual site bound dwell times by randomly assigning each unbinding event to one of the

bound ligands. Unbound dwell times for individual sites were assumed to be on average four times longer than were observed at tetrameric channels with four sites.

We first simulated single-site bound time series by randomly drawing from the single-site dwell time and event amplitude distributions described above, followed by the addition of Gaussian noise to individual event segments. Gaussian noise for individual events was assigned based on the observed linear correlation between event intensity and the standard deviation of within-event fluctuations for events with a duration longer than 10 frames. No noise was added to the baseline at this point. To simulate tetramers comprised of independent CNBDs, we added four single-site time series together, and finally added Gaussian noise to the remaining baseline points based on the average standard deviation of baseline fluctuations in our data. This procedure sums the noise from each bound ligand without quadrupling the baseline noise. In total, we simulated ~60 h of binding data at multiple fcGMP concentrations to obtain a simulated data set similar in size to our experimental data set.

**HMM analysis**. All models were optimized in QuB[47,48] to maximize their likelihood for ~60 h of idealized binding events across all molecules and concentrations. For single-site models (Supplementary Fig. 9) we restricted our analysis to isolated binding events by removing periods with two or more bound ligands, thereby splitting those time series into segments comprised only of singly-bound events. For models of the first and second binding steps (Supplementary Fig. 10), we similarly restricted our analysis to periods with up to two occupied sites by removing bound periods with more than two bound ligands. Dead time in all cases was one frame. Optimized rate constants and their estimated uncertainty are given in Supplementary Tables 2 and 3. The models were ranked according to their relative Bayesian Information Criterion (ΔBIC = BIC − BIC_best model) scores (smaller is better) (Supplementary Figs. 9, 10; and Supplementary Tables 2, 3).

**Reporting summary**. Further information on research design is available in the Nature Research Reporting Summary linked to this article.

## Data availability
The data that support this study are available from the corresponding author upon reasonable request. Source data are provided with this paper.

## Code availability
Data analysis was performed using custom-written scripts in MATLAB available as Supplementary Software 1 and at Github (https://github.com/marcel-goldschen-ohm/single-molecule-imaging-toolbox)[65].

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

## Acknowledgements
We thank Drs. Jonathan Pierce and Iku Mori for gifting cDNA for wild-type TAX-4. We thank Drs. Richard Aldrich and Eric Senning for helpful discussion and shared use of equipment and Dr. Eric Senning for gifting the TRPV1-GFP construct.

## Author contributions
V.R.P. performed molecular biology including generating cDNA for constructs, carried out the experiment, analyzed the data, and contributed to the writing of the manuscript. A.M.S. and D.Q. contributed to data analysis and writing of the manuscript. S.G. performed molecular biology including generating cDNA for constructs as well as all cell culture for experiments. D.J.S. performed preliminary vesicle experiments. M.P.G.-O. conceived, designed, and oversaw the project, and contributed to data analysis and writing of the manuscript.

## Competing interests
The authors declare no competing interests.
