## [Peer Review File · Nature Communications]

Single-molecule imaging with cell-derived nanovesicles reveals early binding dynamics at a cyclic nucleotide-gated ion channelReviewers' Comments:

Reviewer #1:

Remarks to the Author:

The manuscript by Patel et al. uses a powerful technology where single binding events of a fluorescent cGMP to a tetrameric cyclic nucleotide modulated channel (TAX-4) are measured in a quest to gain insights into the channel gating mechanism. The technique is well described, and the single-molecule binding data presented is beautiful, with well resolved single-level binding events obtained using TIRF microscopy, further enhanced with colocalization data from a GFP tag attached to the channel. The use of cell-derived vesicles is relatively new and powerful, as it allows measurements from membrane proteins in a native environment, which is a step further from the authors' previous publication, which uses only the truncated cyclic nucleotide binding domains. The authors used this approach to acquire single-molecule data for the first one-two ligand binding events to TAX4 and then used it to try to discriminate between different gating models reduced to describing how TAX4 binds the first 2 cGMP molecules.

Major concerns:

My main objection with the conclusions in this manuscript is related to the degree to which mechanistic information can be obtained from the models/data presented and how any of these models relate to a realistic picture of how these channels gate in response to ligand binding.

One of the major limitations of this approach is that, as presented and as the authors also state, it is applicable only to reactions where the ligand has quite high binding affinity to the protein (considerably higher than the μM affinity of cGMP binding to TAX4 used here) because at high concentrations of fluorescent ligand the background becomes too high to resolve individual binding events. In this case for example, the authors cannot use sufficiently high concentrations of fcGMP to cover the entire binding curve (Fig.4D). The problem with this is that the data they were able to acquire were extremely limited and a full concentration-dependence set was not determined so that gating models can be properly constrained.

The analysis of all the dwell-time distributions for all ligand concentrations tested shown in Fig 5 yielded 2 exponential components for both bound and unbound states. The authors first fit these data to single binding site models (supp 9). The authors should clarify here whether all the data was used for fitting to these single-binding models, or only data with single-binding events (excluding any stacked events). The mechanistic interpretations derived from fitting only single-binding events with such models would be important and this part should yield interesting novel insights for the full-length channels: having two unbound states and 2 bound states (judging from the bound distributions at 10 and 30 nM fcGMP), as it was previously shown for the soluble CNBDs.

On the other hand, the mechanistic implications of using the two binding site models are less clear. First, although the cartoon in suppl 10a shows that there are two ways to bind the second ligand once the first is already bound, this feature did not make it into the models shown in c. Despite missing this important feature, the authors make strong statements about things like cooperativity using such incomplete models. However, I believe that efforts to understand anything about gating beyond the first binding step from such data as presented here is not realistic. There is not enough data to constrain even the models they show in suppl 10c, so adding another couple of states will not help. I believe that any conclusions drawn based on these 2 binding site models is unwarranted and not supported by their data.

Other comments:

1. Could the authors explicitly outline the expected kinetic differences in photobleaching versus cGMP dissociation and how these are distinguished in their analyses?

2. Could the authors specifically state whether the GFP bleaching to count the subunits is performed for each and every one of the spots they analyze for the single binding events?
3. The authors plotted, in fig 5c, "weighted time constants from biexponential fits of bound and unbound dwell times" but they refer to it in the text as "bound lifetimes reflect the rate of complex dissociation independent of concentration...." (pg 9 bottom). Please clarify why you equate the weighted time constants with the inverse of rate of dissociation/formation. Which model did you use? Wouldn't you rather plot each of the components separately?
4. Fig 4b occasionally shows 3 stacked events. Specify if they are ignored or considered noise.
5. Discuss the missed events and your bandwidth limitations (page 10).
6. Fig. 5d is not very clear. Indications of how the graph will look like in the absence of correlation would help.
7. In general, the models are not particularly well-described (A1, 2, B1, 2, correspond to? pg 12, alpha and beta in suppl tables, constraints mentioned pg 14 but not clarified in these tables). Also, the ranking method is not clear, and what does a deltaBIC score of 0 mean? However, since this part may get rehauled, I will not go into too many details.
8. The discussion is replete with claims not supported by the data, as stated in major concerns. E.g. discussions about dimer of dimers, negative cooperativity, the capping of the binding site with C-helix and movement of CNBD toward membrane, arguments against MWC, are not supported by the data. Discussion section needs major changes.
9. There are no error bars in Fig.3b, 4c, d, 5c
10. Komatsu et al, fig legend of fig 4 is not in the reference list.
11. It is unclear what Fig 6 is meant to convey. Are the authors suggesting that the models in b and c are equally likely? If not, perhaps only the preferred one should be in this figure?

Reviewer #2:

Remarks to the Author:

The authors utilize a combination of single molecule and protein isolation techniques to monitor ligand-ion channel binding events and monitor subsequent changes in ion channel conformation. Membrane receptors are notoriously difficult to image at the single molecule level as they require their "native membrane" to be fully functional and cell based measurements are complicated by a wide range of issues. The authors solve this through their use of cell derived vesicles and single molecule imaging. This is a clever approach. This work is a clear advancement in the application of single molecule techniques for membrane receptors. A few questions, comments, and concerns are listed below.

1. The authors should cite TJ Ha's SIMPul papers along with the cell derived vesicle single molecule work they cite.
2. On line 1 of page 13, the authors use the term widefield imaging to refer to their TIRF measurements. While I understand the point they are trying to make, TIRF is not technically widefield. I would recommend rewording this for clarity.
3. It's not clear how the authors were sure that the fcGMP was not photobleaching rather than unbinding from the ion channel. Something to validate the photostability of these fluorophores or an explanation based on the distribution of events/ statistical analysis is necessary.
4. I read this sentence 5 times before I figured out what they were trying to say. "To determine whether 9 these different conformations arise from distinct populations of molecules or reflect 10 conformational exchange within molecules we examined correlations between the lifetimes of 11 successive binding events at individual molecules (Fig. 5d)." A comma would help.
5. I'm finding it difficult to understand what the authors are trying to say here "The binding cooperativity we are referring to is that 7 between successive binding steps only, irrespective of any cooperativity implied by a global 8 conformational exchange of the CNBDs."
6. Is this statement based on data. "However, at higher concentrations the binomial fits tended to 3 underestimate occupation probability, which could reflect cooperative interactions between 4 binding sites." The authors should explain or point to the figure illustrating this.

7. This is too strong of a statement. "Our results reveal that binding is followed by a conformational change of the bound complex which we discuss below." I agree the most obvious culprit would be a conformational change particularly given the cryoem and FRET literature cited. However, the authors have not directly measured a conformation change. Using the word "likely" would be ok. I hop they plan on extending this approach to smFRET in the future.
8. The authors should include the excitation powers used for their studies.
9. What are the lengths (in time) of the magenta time traces in figure 2?
10. Are the time traces shown in figures 2 and 4 denoised or raw?

Reviewer #1

The manuscript by Patel et al. uses a powerful technology where single binding events of a fluorescent cGMP to a tetrameric cyclic nucleotide modulated channel (TAX-4) are measured in a quest to gain insights into the channel gating mechanism. The technique is well described, and the single-molecule binding data presented is beautiful, with well resolved single-level binding events obtained using TIRF microscopy, further enhanced with colocalization data from a GFP tag attached to the channel. The use of cell-derived vesicles is relatively new and powerful, as it allows measurements from membrane proteins in a native environment, which is a step further from the authors' previous publication, which uses only the truncated cyclic nucleotide binding domains. The authors used this approach to acquire single-molecule data for the first one-two ligand binding events to TAX4 and then used it to try to discriminate between different gating models reduced to describing how TAX4 binds the first 2 cGMP molecules.

Major concerns:

My main objection with the conclusions in this manuscript is related to the degree to which mechanistic information can be obtained from the models/data presented and how any of these models relate to a realistic picture of how these channels gate in response to ligand binding.

One of the major limitations of this approach is that, as presented and as the authors also state, it is applicable only to reactions where the ligand has quite high binding affinity to the protein (considerably higher than the μM affinity of cGMP binding to TAX4 used here) because at high concentrations of fluorescent ligand the background becomes too high to resolve individual binding events. In this case for example, the authors cannot use sufficiently high concentrations of fcGMP to cover the entire binding curve (Fig.4D). The problem with this is that the data they were able to acquire were extremely limited and a full concentration-dependence set was not determined so that gating models can be properly constrained.

We agree with the reviewer that our inability to saturate the binding site using the current approach is a limitation of the current work. Nonetheless, unlike macroscopic ensemble-averaged measures, we observed binding of the first two ligands at single event resolution which resolves sequential binding events unobscured by bulk-averaging and allows us to directly probe differences in binding of the 1st and 2nd ligand. This is a very important point that differentiates our data from macroscopic dose-response current recordings which typically do require the full dose-response relation to reliably model. Indeed, despite not having a full concentration-dependence set, we would argue that our observations are potentially the best available constraint on the dynamics of association for the first two ligands to date. First, they reveal that binding of the first ligand to the full-length channel is followed by a conformational change of the CNBD similar to that observed in isolated purified CNBDs from HCN2 channels, and that this conformational change inhibits association/dissociation from the site. Also, no amount of higher concentration data will change the fact as revealed by our observations that binding of the 2nd ligand is not enormously different than binding of the 1st. Although this does not directly reveal the mechanism of ligand-activation, it has considerable bearing on it. For example, a previous

study proposed that binding of the 2nd ligand in CNGA2 channels is orders of magnitude less favorable than the first, due to the 2nd binding event being associated with the energetic cost of pore opening (PMID 17322905). For TAX-4 at least, our data suggest that this mechanism is not plausible. Also, although limited in concentration, our dataset included ~60 hours of binding dynamics, which is much larger than typical single molecule datasets.

The analysis of all the dwell-time distributions for all ligand concentrations tested shown in Fig 5 yielded 2 exponential components for both bound and unbound states. The authors first fit these data to single binding site models (supp 9). The authors should clarify here whether all the data was used for fitting to these single-binding models, or only data with single-binding events (excluding any stacked events). The mechanistic interpretations derived from fitting only single-binding events with such models would be important and this part should yield interesting novel insights for the full-length channels: having two unbound states and 2 bound states (judging from the bound distributions at 10 and 30 nM fcGMP), as it was previously shown for the soluble CNBDs.

We did state in the methods that the single-site models were constrained with data comprised only of isolated single binding events (page 29, line 6) as well as in the caption of Supp Fig. 9. To make this more apparent, we now additionally clarify this in the main text (page 11, line 7). We agree with the reviewer that these data offer important mechanistic insights which we discuss in the 2nd paragraph of the discussion section.

On the other hand, the mechanistic implications of using the two binding site models are less clear. First, although the cartoon in suppl 10a shows that there are two ways to bind the second ligand once the first is already bound, this feature did not make it into the models shown in c. Despite missing this important feature, the authors make strong statements about things like cooperativity using such incomplete models. However, I believe that efforts to understand anything about gating beyond the first binding step from such data as presented here is not realistic. There is not enough data to constrain even the models they show in suppl 10c, so adding another couple of states will not help. I believe that any conclusions drawn based on these 2 binding site models is unwarranted and not supported by their data.

One thing is very clear regarding the two-site models: A simple sequential two-step binding model is insufficient to explain our observations of fcGMP association even if the dynamics of the two steps are allowed to be cooperative (compare BIC scores in Supp Fig 10b and simulated dwell time distributions in Supp Fig 11 for models M2.Ai/c with all other M2 models). To explore the question of whether differences between adjacent vs. diagonally occupied sites contribute to observations of doubly bound events we have added a model that treats these conformations as distinct di-liganded states (M2.B in Supp Fig. 10; discussed on page 12, line 19). Note that the letter identifier for the rest of the models has been shifted from B-D to C-E. This model is only slightly better than the simplest two-site models and ranks much worse than any of the models that postulate a conformational change within ligation states (M2.C-E). Thus, although we certainly cannot rule out that orientation of occupied sites in the tetramer may be a factor in determining binding dynamics, it appears for the first two binding events that a conformational change separate from association/dissociation is an important feature. Thus, we

explored two-site models M2.C-E that postulate either a global or per subunit conformational change of the binding domain following ligand association as per our observations for single-site dynamics. Overall, we agree with the reviewer that we were unable to determine whether the conformational change was global or occurred independently in each subunit, a fact that we were up front about in the text (page 14, line 13; page 19, line 15). It is also true that we did not include distinct doubly bound states for occupation of either adjacent or diagonally opposite sites in models M2.C-E as this would have made them overly complex. Although further examination of geometry dependent binding is certainly of interest, we note that the single molecule binding dynamics reported here suggest that a conformational change as observed at single sites is an important feature of the observed dynamics irrespective of whether distinct binding geometry may also play a role. We note that differential binding in adjacent versus diagonal sites cannot explain the multiple bound conformations at individual sites. We have amended the text to discuss these points (pages 12-13). Our conclusions based on cooperativity of the 2nd ligand stem from a combination of direct experimental observations that dynamics in singly and double bound states differ only subtly from that expected for independent binding sites (Supp Figs. 12-13) and uniform agreement across all tested models that any cooperativity in the 2nd binding equilibrium constant should be less than an order of magnitude. These conclusions are based on ~60 hours of single-molecule binding dynamics across concentrations that shift occupancy primarily between 0-2 ligands. These data provide an unprecedented, detailed look at distinct dynamics in singly and doubly bound states not directly attainable in ensemble-averaged measures. We acknowledge that overly complex HMM models are likely to provide nonunique solution parameters with little relevance to actual mechanisms. However, models of the complexity explored here are routinely used to describe ion channel function. Furthermore, we have shown that different folds of the dataset provide nearly identical solutions with small deviations in parameters (Supp Tables 2-3). Also, all of the models consistently suggest that a conformational change is a necessary feature to explain the underlying dynamics, even though we have now added an additional model that treats distinct geometrical diliganded states (e.g. adjacent vs. diagonal; M2.B in Supp Fig. 10b). This same observation is the conclusion at individual sites and conforms with our prior observations for isolated CNBDs. Thus, we would kindly ask the reviewer to clarify their reasoning and/or the metric on which they base their opinion that we do not have enough data to constrain the models in Supp Fig. 10c and their strong statement that any conclusions drawn from these models are unwarranted and not supported by the data.

Other comments:

1. Could the authors explicitly outline the expected kinetic differences in photobleaching versus cGMP dissociation and how these are distinguished in their analyses?

We agree that this is an important point. Please see our response to comment #3 from Reviewer #2 who also raised this issue.

2. Could the authors specifically state whether the GFP bleaching to count the subunits is performed for each and every one of the spots they analyze for the single binding events?

GFP bleach steps were evaluated for each and every colocalized spot, see text (page 8, line 21) and methods. We have additionally mentioned this for clarity (page 25, line 6).

3. The authors plotted, in fig 5c, "weighted time constants from biexponential fits of bound and unbound dwell times" but they refer to it in the text as "bound lifetimes reflect the rate of complex dissociation independent of concentration..." (pg 9 bottom). **Please clarify why you equivate the weighted time constants with the inverse of rate of dissociation/formation.** Which model did you use? Wouldn't you rather plot each of the components separately?

We apologize for the confusion as we did not intend to equivate weighted time constants with inverse rates of dissociation/formation. Our intention was merely to indicate that our observations were qualitatively in line with that expected for a simple binding reaction where one should expect unbound durations to depend on ligand concentration and bound durations to be independent of concentration. We have amended the text to clarify this by removing statements about specific rates (page 9, line 21).

4. Fig 4b occasionally shows 3 stacked events. Specify if they are ignored or considered noise.

Our idealization procedure did not specifically ignore events with more than two occupied binding sites. See Supp Fig. 7 for an example where such events are identified in the idealization. In the case of the brief spikes seen in the trace shown in Fig. 4b, these were identified as noise rather than a third ligand. However, we do report that our ability to resolve single frame events as validated by simulation is reduced for stacked events as compared to isolated events, and likely to be even poorer for triply bound events lasting only a single frame (see Supp Fig. 8). Thus, it is possible that the brief spikes seen in the trace shown in Fig. 4b are missed triply bound single frame events, although given the additive noise of multiple ligands they may also simply be noise as determined during idealization. We have added statements to this effect in the text for clarity (page 27, last paragraph).

5. Discuss the missed events and your bandwidth limitations (page 10).

We evaluated the ability of our idealization procedure to detect binding events using simulated data with dwell times and noise drawn from our experimental observations (see methods page 27, last paragraph, and Supp Figs. 7-8). Supp Fig. 8 depicts the ability of this procedure to identify singly and doubly liganded events as a function of their duration and indicates a very good match with known bound durations except for brief single frame events which are predominantly missed for dwells in doubly liganded states. However, in all cases the match is very good for events lasting two or more frames. In addition to the previous discussion of this in the methods, we have added a discussion of this to the main text (page 10, line 3).

6. Fig. 5d is not very clear. Indications of how the graph will look like in the absence of correlation would help.

We have added a dashed diagonal line to the plots and associated description in the figure caption and main text (page 10, line 18) to clarify this.

7. In general, the models are not particularly well-described (A1, 2, B1, 2, correspond to? pg 12, alpha and beta in suppl tables, constraints mentioned pg 14 but not clarified in these tables). Also, the ranking method is not clear, and what does a deltaBIC score of 0 mean? However, since this part may get rehailed, I will not go into too many details.

We redefined the model labels as M1.* (one site models) and M2.* (two site models) rather than the previous M01.* and M012.* for simplicity. We also redefined the label indicating the model constraints so as to be more intuitive (M2.Ai and M2.Ac refer to model structure M2.A with independent or cooperative binding steps, respectively, as opposed to the previous A1 and A2). We have added a much more thorough explanation of the models including their labels and various constraints in the captions of Supp Fig. 10 and Supp Table 3. We now define deltaBIC in the text (page 11, line 11 and page 29, line 13) and in the captions of Supp Figs 9-10. Finally, we also rearranged the panels in Supp Figs. 9-10 so the flow proceeds from models to BIC scores rather than the reverse (however, content is unchanged).

8. The discussion is replete with claims not supported by the data, as stated in major concerns. E.g. discussions about dimer of dimers, negative cooperativity, the capping of the binding site with C-helix and movement of CNBD toward membrane, arguments against MWC, are not supported by the data. Discussion section needs major changes.

All the points listed above other than arguments against MWC are references to conclusions from prior studies which we use to help frame the discussion of our data, not direct conclusions from the data in this manuscript. We do not feel that we have written the discussion in such a way that would be misleading and suggest that these ideas are based on our own results. For example, our prior discussion of the coupled dimer model was if anything an admission that we cannot say much about this model one way or the other rather than any claim regarding it (although we have removed some of the prior text regarding this, the remaining text still makes this clear; page 19, line 21). For capping of the binding site by the C helix, we specifically state that we previously postulated this mechanism based on prior structures of apo and holo isolated CNBDs and our previous measurements of binding dynamics as isolated CNBDs from HNC2 channels (page 18, line 4), and that it is also consistent with previous measures of FRET and DEER at CNBDs (page 18, line 8). We have recited our previous work (page 18, line 7) and added additional text in various places (e.g. page 11, line 22) to help clarify this, although in general we feel that we have not been remiss in attaching citations to these references. We also cite cryo-EM structures and other functional evidence such as high-speed AFM regarding motions of the CNBDs toward the lipid membrane (page 18, line 15). Regarding the MWC mechanism (M2.D) for binding (not pore gating), given that this full cyclic model predicts that the conformational change inhibits binding, we favored the simpler model M2.C where binding is disallowed following the conformational change. First, we have amended the text to make it clearer that rather than ruling out an MWC mechanism, we find that it suggests a simpler mechanism which was nearly equally likely based on our data, and thus we favored the simpler model (page 14, line 1). Second, we have removed most of the discussion of the MWC model

from the Discussion section as it did not add much value beyond what we already discussed in the Results. Third, it would be a great help to us if the reviewer could please define what exactly it is about our arguments that they feel are not supported by the data.

9. There are no error bars in Fig.3b, 4c, d, 5c

The distributions in Figs. 3b and 4c are simply the raw data distributions over all molecules plotted next to their binomial maximum likelihood fits, and thus there are no errorbars. However, we now report the 95% confidence interval for the binomial maximum likelihood fit parameter in the captions. In Figs. 4d and 5c we have added a shaded region denoting either the standard error measure (it is difficult to see without zooming in) or the 95% CI. We also added 95% CI to biexponential parameters in Supp. Table 1.

10. Komatsu et al, fig legend of fig 4 is not in the reference list.

Thank you for catching this oversight. We have added it.

11. It is unclear what Fig 6 is meant to convey. Are the authors suggesting that the models in b and c are equally likely? If not, perhaps only the preferred one should be in this figure?

We apologize for the confusion. It was indeed our intent to suggest that we cannot distinguish between a global vs. per subunit conformational change after binding. We have amended the caption to clarify this.

Reviewer #2

The authors utilize a combination of single molecule and protein isolation techniques to monitor ligand-ion channel binding events and monitor subsequent changes in ion channel conformation. Membrane receptors are notoriously difficult to image at the single molecule level as they require their “native membrane” to be fully functional and cell-based measurements are complicated by a wide range of issues. The authors solve this through their use of cell derived vesicles and single molecule imaging. This is a clever approach. This work is a clear advancement in the application of single molecule techniques for membrane receptors. A few questions, comments, and concerns are listed below.

1. The authors should cite TJ Ha’s SIMPul papers along with the cell derived vesicle single molecule work they cite.

Done (page 6, line 6).

2. On line 1 of page 13, the authors use the term widefield imaging to refer to their TIRF measurements. While I understand the point they are trying to make, TIRF is not technically widefield. I would recommend rewording this for clarity.

We agree that this could be confusing and have amended this portion of the text (page 3, line 23).

3. It’s not clear how the authors were sure that the fcGMP was not photobleaching rather than unbinding from the ion channel. Something to validate the photostability of these fluorophores or an explanation based on the distribution of events/ statistical analysis is necessary.

We agree that this is an important point. We previously relied on the observation that noncolocalized fcGMP events that we assume primarily represent bleaching of dye adsorbed onto the surface tend to be much longer-lived than binding events. We have now quantified this qualitative observation with a formal analysis of noncolocalized fcGMP events. We show that noncolocalized events are much longer lived than most binding events, suggesting that the vast majority of observed bound durations are terminated by unbinding rather than bleaching (Supp. Fig. 14). We have also added a methods section detailing this analysis (page 25, line 11).

4. I read this sentence 5 times before I figured out what they were trying to say. “To determine whether these different conformations arise from distinct populations of molecules or reflect conformational exchange within molecules we examined correlations between the lifetimes of successive binding events at individual molecules (Fig. 5d).” A comma would help.

We have amended this sentence for clarity (page 10, line 14).

5. I’m finding it difficult to understand what the authors are trying to say here “The binding cooperativity we are referring to is that between successive binding steps only, irrespective of any cooperativity implied by a global conformational exchange of the CNBDs.”

It was our intent to specify that even with binding steps constrained to be independent, cooperativity could still exist in model M2.D due to differential binding following the

conformational change. However, since we explicitly refer to only models M2.C and M2.E in this instance, the clarification is both unneeded and confusing. Thus, we have removed this sentence.

6. Is this statement based on data. “However, at higher concentrations the binomial fits tended to underestimate occupation probability, which could reflect cooperative interactions between binding sites.” The authors should explain or point to the figure illustrating this.

We now cite the data in Fig. 4c on which this statement is based (page 15, line 1).

7. This is too strong of a statement. “Our results reveal that binding is followed by a conformational change of the bound complex which we discuss below.” I agree the most obvious culprit would be a conformational change particularly given the cryoem and FRET literature cited. However, the authors have not directly measured a conformation change. Using the word “likely” would be ok. I hope they plan on extending this approach to smFRET in the future.

We have amended this sentence by replacing “reveal” with “suggest” in line with the reviewer’s suggestion (page 17, line 20). We also agree that smFRET in combination with nanovesicles is an exciting prospect.

8. The authors should include the excitation powers used for their studies.

We now report the excitation laser power at the sample in the methods (page 24, line 4).

9. What are the lengths (in time) of the magenta time traces in figure 2?

We have corrected the aberrant scale bars in Fig. 2a and apologize for the confusion this caused. Each of the three traces below each image is 100 seconds long.

10. Are the time traces shown in figures 2 and 4 denoised or raw?

The time traces shown in Figs. 2 and 4 are the raw time series z-projections through the recorded image stack of the average intensity within a radius of 2.5 pixels of the identified spot central pixel. No denoising was applied.

Reviewers' Comments:

Reviewer #1:

Remarks to the Author:

In the revised manuscript, the authors have addressed a few of my concerns. The assessment of an additional model where there are 2 possibilities for the second binding event (adjacent and diagonal) is a nice addition. They added needed details about the methods, clarified the models and some of the text.

On the other hand, major concerns remain, still related to the conclusions drawn from the models that include a second ligand-binding site meant to approximate ligand binding and conformational changes in TAX4 from single and double ligand-binding steps. The authors use ~5 different models (Suppl fig 10) to fit their data in order to extract values for the binding rate constants to assess cooperativity. They rank these models using the Bayesian information criterion (BIC).

The first major issue is related to the fitted data. The second binding level introduces uncertainties in the analysis because once the second ligand is bound, the durations of the individual events are unknown. Transition to the first level from the second level means that either the first ligand or the second ligand unbound, and an error in assignment here, leads to errors in assigning all events in a burst (e.g. such as that seen in Fig 4 at high ligand concentrations). The authors assign the events randomly to handle this issue and justify this approach by stating that the dwell-time distributions are not very different when different random assignments are made (analysis not shown?). However, these are exactly the data that would speak to whether cooperativity is at play or not. The authors need to justify the validity of this particular approach.

The second major issue is related to the type of information garnered from fitting the data with the different models. The two simple sequential models that do not include additional states (M2A,B) have the worst scores and are discarded. This is not surprising, since they already showed that the single binding event data needed 2 components to fit the unbound dwell-times and 2 components for the bound, as already seen in the analysis of the single-binding events data, so the simple sequential models will not work (M2A). The authors pick one of the 6 state models, M2C, and argue that releasing the independence constraint in the ligand binding rates does not lead to a model where the best-fit rates are that different from the independent case, and thus, they conclude that there is no evidence of strong cooperativity. However, upon addition of the needed extra state for each bound conformation (the 6 state models), the models become too complex in that the number of parameters in the models (M2C, D, E have upwards of 10 parameters each) exceeds that of the parameters extracted from the data and the fits become poorly constrained. Proof in point, the 6 state models have similar rankings (according to the authors, page 13-14, although a significance level for the score is not provided. Is a difference of 3000 as in M2E significant, where is the line drawn?). In consequence, I am not convinced from the analysis shown whether the rates they list in suppl Table 3 are unique and thus whether their conclusion is warranted. The authors may want to show that models with neither negative cooperativity nor positive cooperativity (by constraining those rates accordingly) can fit the data. In addition, this analysis may need to be performed on M2C and M2D, as well as M2E since the authors kept Fig.7c as a potential viable model.

The third major issue is related to the concept of cooperativity. The authors should be more clear in the text regarding the type of cooperativity they are talking about. Initially, given how the models were set up, I thought that by cooperativity, they meant strictly a comparison between the equilibrium constants of the intrinsic binding (0-1, 1-2). However, in the text they refer frequently to cooperativity of activation, especially when they compare their data with other reports (where ligand binding and channel opening at concomitantly measured), and it becomes confusing. They should define this clearly and specifically in the text where it first appears.

In addition, the reason for rejecting model 2D is not clear. It sounds as if when ligands bind, the

conformational exchange becomes slower? or less likely? Or both? It almost sounds like negative cooperativity with respect to the conformational change, which the authors propose is related to activation. Is the starred conformation more or less likely to occur in M2D upon successive ligand binding steps? More clarification is needed here.

The authors should also include equilibrium constants for the binding transitions rather than only the on and off rate constants in tables such as the Suppl table 3

Page 19 discussion. The authors have no data that speak to the likelihood of a dimer of dimers model. Also, in depth structural hypotheses such as connecting the additional state the authors found with a conformation where "CNBD is capped by the C helix combined with a movement towards the membrane" are a bit farfetched.

The authors state that models M2C and D have similar rankings (differ by about 1000), however, the authors do not give a level of significance. What score difference is considered significant?

Reviewer #2:

Remarks to the Author:

The authors have adequately addressed my concerns.

REVIEWER 1

In the revised manuscript, the authors have addressed a few of my concerns. The assessment of an additional model where there are 2 possibilities for the second binding event (adjacent and diagonal) is a nice addition. They added needed details about the methods, clarified the models and some of the text.

On the other hand, major concerns remain, still related to the conclusions drawn from the models that include a second ligand-binding site meant to approximate ligand binding and conformational changes in TAX4 from single and double ligand-binding steps. The authors use ~5 different models (Suppl fig 10) to fit their data in order to extract values for the binding rate constants to assess cooperativity. They rank these models using the Bayesian information criterion (BIC).

We thank the reviewer for their helpful comments and believe that our revised manuscript is much improved for them. We have reworked several sections of the text primarily regarding the 2nd binding step and cooperativity in the results and the discussion to address the reviewer's comments, clarify our point of view, and to improve the flow of the text. We also note that in addition to our responses below we have added a new Fig. S5 which addresses colocalization of binding with TAX-4 in comparison to that for a TRPV1 channel as a control. We feel that this control is a bit better than that with GABAA receptors as the GFP tag on TRPV1 was also on the intracellular side of the membrane analogous to TAX-4.

We also apologize for the apparent tardiness of this response. However, we did not receive the reviewer's comments from the journal until September 3rd.

New Items:

- Fig. 6d
- Fig. S5
- Fig. S15
- Table S4
- Table S5
- Methods section on dwell time distributions

Updated Items:

- Fig. S9c
- Fig. S10b-c
- Table S2
- Table S3

The first major issue is related to the fitted data. The second binding level introduces uncertainties in the analysis because once the second ligand is bound, the durations of the individual events are unknown. Transition to the first level from the second level means that either the first ligand or the second ligand unbound, and an error in assignment here, leads to errors in assigning all events in a burst (e.g. such as that seen in Fig 4 at high ligand concentrations). The authors assign the events randomly to handle this issue and justify this approach by stating that the dwell-time distributions are not very different when different random assignments are made (analysis not shown?). However, these are exactly the data that would speak to whether cooperativity is at play or not. The authors need to justify the validity of this particular approach.

We acknowledge that we do not determine which of the two bound ligands unbinds during unbinding from stacked events. However, it is not essential to know which ligand unbound to assess cooperativity. Cooperativity is not a measure of the bound duration of individual ligands regardless of their binding order, but rather a measure of the time spent in doubly liganded vs singly liganded states, which we do resolve. Thus, despite the ambiguity in which ligand unbinds, we feel that we are well within our means to evaluate cooperativity based on our data. We now mention this explicitly in our revised results section on binding cooperativity.

The bound duration distributions were used solely as a guide to develop reasonable models. Namely, their biexponential nature suggested models with at least two distinct bound states. That said, we acknowledge that the generation of dwell time distributions by random assignment of which ligand unbinds from stacked events leaves some uncertainty in the validity of the resulting distributions. To address this, we have re-generated these distributions under the two extreme assumptions that the first ligand to bind is always either the first or the last to unbind (new Fig. S15). Although these assumptions result in different proportions of short vs. long dwells, they do not change the overall observation that the bound distributions are at least biexponential (i.e. binding sites adopt at least two distinct bound states). Given that the description of the computation for the dwell times was clearly disruptive to the flow of the main text, we have moved it to the Methods where it has an expanded discussion in its own section.

The second major issue is related to the type of information garnered from fitting the data with the different models. The two simple sequential models that do not include additional states (M2A,B) have the worst scores and are discarded. This is not surprising, since they already showed that the single binding event data needed 2 components to fit the unbound dwell-times and 2 components for the bound, as already seen in the analysis of the single-binding events data, so the simple sequential models will not work (M2A).

We agree with the reviewer that it is not very surprising that model M2.Ai (sequential binding to independent sites) does not work. Although perhaps not a huge surprise, it was not immediately obvious to us that allowing the 2nd binding step to differ from the 1st (M2.Ac) would not have been sufficient, which it is not. In fact, M2.Ac has just as many free parameters (4) as the much more preferred model M2.Fii. Finally, M2.B with two distinct doubly bound states and 6 free parameters also does not work, which we feel was not at all obvious.

The authors pick one of the 6 state models, M2C, and argue that releasing the independence constraint in the ligand binding rates does not lead to a model where the best-fit rates are that different from the independent case, and thus, they conclude that there is no evidence of strong cooperativity. However, upon addition of the needed extra state for each bound conformation (the 6 state models), the models become too complex in that the number of parameters in the models (M2C, D, E have upwards of 10 parameters each) exceeds that of the parameters extracted from the data and the fits become poorly constrained.

First, we wish to point out that all versions of M2.E and M2.F have either 4 or 6 free parameters only (non-shaded entries in Table S3 excluding deltaBIC) and allowing cooperative binding in M2.F has essentially the same effect as in M2.C/D (i.e., a bit better but not much – see our response below for how we determine what “much” means). Since M2.B has 6 free parameters and was much worse in terms of its deltaBIC score, we conclude that having 6 free parameters alone does not result in unidentifiable models in general, and thus it is reasonable to evaluate these models given the data.

That said, we acknowledge that either 8 or 10 free parameters for versions of M2.D is a lot. To address this, we have added an additional model scheme (new M2.C; prior models M2.C-E were renamed to M2.D-F) which is nearly identical to M2.D but lacks the 0* state (i.e., does not allow a conformational exchange in the absence of bound ligand). This simplification is reasonable given that for the single site models the exchange is much slower in the absence of ligand and ignoring it altogether has only a minor impact on BIC score (furthermore, we already ignore this state in M2.F). Removal of the 0* state reduces the number of free parameters for M2.C to 6 (independent sites) or 8 (cooperative sites) and results in rate constants that are very close to those for the M2.D model that includes the 0* state. Thus, the additional 2 free parameters in M2.D do not have a large effect on the other rates in the model and are unlikely to render the result ambiguous. Nonetheless, 8 params is still a fair amount, which we explicitly acknowledge in the revised section on the first two binding steps. Furthermore, as suggested by the reviewer we have also evaluated models with defined cooperativity (see below). Taken together, we still feel that given our data we can make the case that either negative cooperativity or very strong (i.e. 100-fold increase in binding equilibrium constant) positive cooperativity are much less likely than either independent sites or more modest positive binding cooperativity. We feel that the revised text does a better job of explaining what our data and the explored models can and cannot say.

Proof in point, the 6 state models have similar rankings (according to the authors, page 13-14, although a significance level for the score is not provided. Is a difference of 3000 as in M2E significant, where is the line drawn?). In consequence, I am not convinced from the analysis shown whether the rates they list in suppl Table 3 are unique and thus whether their conclusion is warranted.

The reviewer raises an excellent point in that we were cavalier with our discussion of what differences in deltaBIC scores mean. To address this, we have added error bars on all deltaBIC scores which we derived from the standard deviation of BIC scores across 5 separate folds of the dataset (each randomized fold contained ~20% of the molecules at each concentration, each molecule was assigned to only one of the folds). We evaluated significant differences with one-way ANOVA and post-hoc Tukey test at $p < 0.05$. We now report these errors in Tables S2 and S3 and graphically depict them in the bar plots in Fig. 6d and Figs. S9-10.

It is not feasible to evaluate all possible combinations of rate constants, so we cannot say with certainty that the optimized rates are absolutely unique. However, the optimized rate constants are highly consistent within each model across different folds of the dataset (reported errors for rate constants in Table S3 are standard deviations across 5 randomized folds of the dataset), and also generally consistent across models.

The authors may want to show that models with neither negative cooperativity nor positive cooperativity (by constraining those rates accordingly) can fit the data. In addition, this analysis may need to be performed on M2C and M2D, as well as M2E since the authors kept Fig.7c as a potential viable model.

This is an excellent suggestion. We now evaluate 10- or 100-fold cooperativity (both positive and negative) for binding equilibrium constants in models M2.C, M2.D and M2.F (new Fig. 6d; new Table S4). For simplicity we split the change in equilibrium evenly between binding and unbinding rates. We do the same for M2.E but assign the cooperativity to the conformational exchange in the spirit of a MWC model. In all cases, positive cooperativity provided a better description of the data than negative cooperativity and 10-fold was better than 100-fold. Overall, these results suggest that 10-fold positive cooperativity cannot be distinguished from models with variable cooperativity or independent binding, whereas negative cooperativity or 100-fold positive cooperativity provide poorer descriptions of the data.

The third major issue is related to the concept of cooperativity. The authors should be more clear in the text regarding the type of cooperativity they are talking about. Initially, given how the models were set up, I thought that by cooperativity, they meant strictly a comparison between the equilibrium constants of the intrinsic binding (0-1, 1-2). However, in the text they refer frequently to cooperativity of activation, especially when they compare their data with other reports (where ligand binding and channel opening at concomitantly measured), and it becomes confusing. They should define this clearly and specifically in the text where it first appears.

We apologize for the confusion on what we mean by cooperativity. Indeed, we are usually strictly talking about a comparison between the equilibrium constants of the intrinsic binding steps (0-1, 1-2). However, we do also evaluate an MWC model (M2.D) where cooperativity arises from differential affinities in the two global conformations rather than from explicit changes in binding rates within a given conformation. As this model is not one that we prefer, it is only discussed briefly. In our revised text we have attempted to clarify exactly what we mean by binding cooperativity throughout.

In addition, the reason for rejecting model 2D is not clear. It sounds as if when ligands bind, the conformational exchange becomes slower? or less likely? Or both? It almost sounds like negative cooperativity with respect to the conformational change, which the authors propose is related to activation. Is the starred conformation more or less likely to occur in M2D upon successive ligand binding steps? More clarification is needed here.

We apologize for the confusion. The equilibrium constant for the conformational exchange to the starred conformation in M2.E (previously M2.D) is 1.6-fold higher (more likely) upon each sequential binding step. However, both the forward and backward rates for the exchange are slowed by 5-10 fold upon each binding step. It's just that the backward rate is slowed by approximately twice that of the

forward rate (see alpha versus beta in Table S3). Thus, the exchange is still favored upon ligand binding, but ligand also slows its dynamics considerably. By construction in this model, this means that binding and unbinding in the isomerized state are also both slowed. We interpret this as supporting the idea that both binding and unbinding are less frequent following the conformational exchange, and hence we favor the simpler models M2.C/D where these transitions are removed. We do not however wish to imply that the conformational exchange absolutely prevents binding, only that it reduces the frequency of binding/unbinding and that a simpler model excluding these transitions is a reasonable approximation. We also note that for M2.E the conformational exchange rates $1 \leftrightarrow 1^*$ are very slow and far from those observed in the single site models. In contrast, this exchange is close to that observed in single sites for models M2.C, M2.D and M2.F. We feel that the slower kinetics in M2.E are an artifact of the model that stems from the slower association rates after the conformational exchange to which they are linked via alpha and beta parameters, and thus we disfavor this model. We clarify these points in the revised text.

The authors should also include equilibrium constants for the binding transitions rather than only the on and off rate constants in tables such as the Suppl table 3

Although we have run out of space to include more columns in Table S3, we now include an additional Table S5 with equilibrium constants as requested.

Page 19 discussion. The authors have no data that speak to the likelihood of a dimer of dimers model. Also, in depth structural hypotheses such as connecting the additional state the authors found with a conformation where "CNBD is capped by the C helix combined with a movement towards the membrane" are a bit farfetched.

We agree that we do not have data that speaks to a dimer of dimers model, which we previously alluded to but now explicitly acknowledge. We likewise agree that we do not have any data that directly addresses the structural hypothesis presented regarding C helix capping. Nonetheless, we have previously published data for homologous HCN channels using a combination of single-molecule binding and crystal structures which do provide evidence for this admittedly highly simplified idea. Thus, we feel that it is relevant to discuss this hypothesis given that our current data are consistent with it. That said, our revised text clarifies that whereas our data are consistent with this hypothesis, they do not provide direct evidence for it.

The authors state that models M2C and D have similar rankings (differ by about 1000), however, the authors do not give a level of significance. What score difference is considered significant?

Copy of response from above:

The reviewer raises an excellent point in that we were cavalier with our discussion of what differences in Δ BIC scores mean. To address this, we have added error bars on all Δ BIC scores which we derived from the standard deviation of BIC scores across 5 separate folds of the dataset (each fold contained ~20% of the molecules at each concentration, each molecule was assigned to only one of the folds). We evaluated significant differences with one-way ANOVA and post-hoc Tukey test at $p < 0.05$. We now report these errors in Tables S2 and S3 and graphically depict them in the bar plots in Fig. 6d and Figs. S9-10.